# COLLABLLM: From Passive Responders to Active Collaborators

**Shirley Wu**[1]  **Michel Galley**[2]  **Baolin Peng**[2]  **Hao Cheng**[2]  **Gavin Li**[1]  **Yao Dou**[3]  **Weixin Cai**[1]
**James Zou**[1]  **Jure Leskovec**[1]  **Jianfeng Gao**[2]

http://aka.ms/CollabLLM

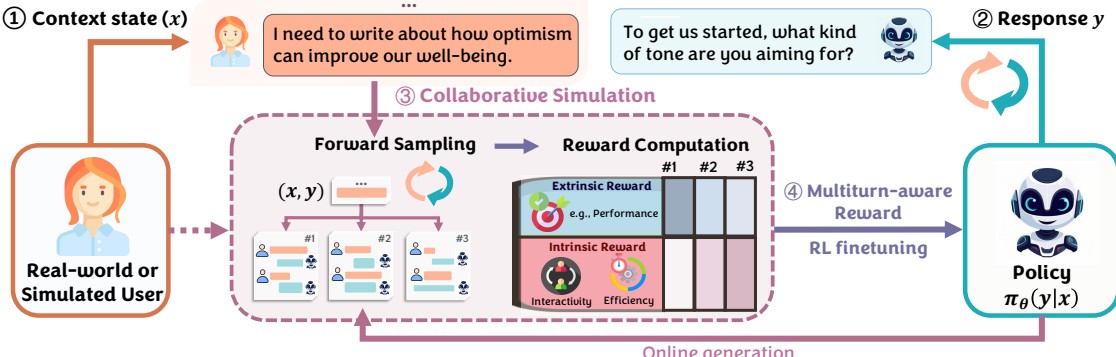

Figure 1: COLLABLLM Framework: Given a context ①, the model generates a response ② to maximize long-term collaboration gains, termed *Multiturn-aware Rewards* (MR). During training, MRs are estimated via ③ collaborative simulation, which forward-samples conversations with simulated users. Finally, ④ reinforcement fine-tuning is applied using the MRs.

## Abstract

Large Language Models are typically trained with next-turn rewards, limiting their ability to optimize for long-term interaction. As a result, they often respond passively to ambiguous or open-ended user requests, failing to help users reach their ultimate intents and leading to inefficient conversations. To address these limitations, we introduce COLLABLLM, a novel and general training framework that enhances multiturn human-LLM collaboration. Its key innovation is a collaborative simulation that estimates the long-term contribution of responses using *Multiturn-aware Rewards*. By reinforcement fine-tuning these rewards, COLLABLLM goes beyond responding to user requests, and actively uncovers user intent and offers insightful suggestions—a key step towards more human-centered AI. We also devise a multiturn interaction benchmark with three challenging tasks such as document creation. COLLABLLM significantly outperforms our baselines with averages of 18.5% higher task performance and 46.3%

improved interactivity by LLM judges. Finally, we conduct a large user study with 201 judges, where COLLABLLM increases user satisfaction by 17.6% and reduces user spent time by 10.4%

## 1. Introduction

Modern Large Language Models (LLMs) excel at generating high-quality single-turn responses when given well-specified inputs. However, real-world users often do not fully articulate their intents and sometimes initiate conversations with an imprecise understanding of their own needs (Taylor, 1968). As a result, users routinely refine their requests post hoc through iterative corrections, which can increase frustration, hinder effective task completion, and reduce conversational efficiency (Amershi et al., 2019; Zamfirescu-Pereira et al., 2023; Wang et al., 2024; Kim et al., 2024). Therefore, an open problem is to train models that actively guide users in clarifying and refining their intents, and helps them achieve their goals. This key challenge would improve user satisfaction and efficiency and streamline human-LLM interactions—especially as LLMs are being applied to real-world tasks that are increasingly complex and open-ended.

A notable limitation of established fine-tuning techniques, such as Reinforcement Learning from Human Feedback (RLHF) (Ouyang et al., 2022), is that they primarily reward LLMs for immediate, single-turn responses, reducing their incentive to seek clarification or assist users in refin-

---

[1]Stanford University [2]Microsoft [3]Georgia Tech. Correspondence to: <shirwu@stanford.edu>, <mgalley@microsoft.com>.

*Proceedings of the 42nd International Conference on Machine Learning*, Vancouver, Canada. PMLR 267, 2025. Copyright 2025 by the author(s).

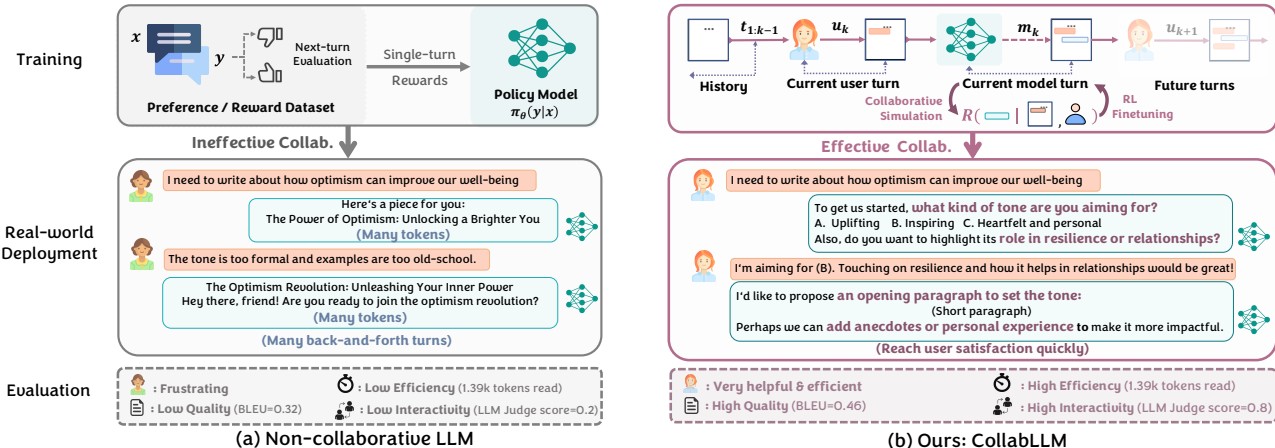

Figure 2: Real examples from COLLABLLM and non-collaborative LLMs. (a) Non-collaborative LLM fine-tuing relies single-turn rewards on immediate responses, which exhibits passive behaviors that follow the user's requests, leading to user frustration, less efficient process, and less satisfactory results. (b) COLLABLLM incorporates Multiturn-aware Rewards from collaborative simulation, enabling forward-looking strategies. This results in more high-performing, efficient, and interactive conversations that anticipate future needs, propose timely clarification, and provide insightful suggestions.

ing their intents or preferences. As a result, commonly used LLMs tend to prioritize direct answers, even though seeking additional context would enhance task completion and increase user satisfaction (Kim et al., 2024).

Here we introduce **COLLABLLM**, a novel and general training framework that improves the ability of LLMs to effectively collaborate with humans in multiturn scenarios (Gao et al., 2019; Balog & Zhai, 2023; Rahmani et al., 2023). The key innovation of COLLABLLM is to promote LLMs' forward-looking behavior that leads to long-term collaboration gains (Figure 1). We introduce a collaborative simulation module that samples future conversations with users to estimate the long-term impact of model responses across multiple turns, a measure we term the *Multiturn-aware Reward (MR)*. The MR function evaluates responses by incorporating both extrinsic metrics, such as task-specific success, and intrinsic metrics, such as efficiency, to holistically assess collaboration quality (*cf.* Section 3). By fine-tuning with RL algorithms (Rafailov et al., 2023; Schulman et al., 2017) on MRs, COLLABLLM promotes responses that lead to better task completion and efficiency in later conversation stages. As shown in Figure 2b, the fine-tuned model goes beyond simply responding to user requests in Figure 2a—it actively collaborates by asking follow-up questions about the writing tone, generating targeted content about the role of optimism, and offering insightful suggestions such as adding anecdotes.

We also introduce **three challenging multiturn tasks** for training and evaluation in simulated environments: MediumDocEdit-Chat, BigCodeBench-Chat, and MATH-Chat, which respectively encompass document creation, code generation, and multiturn question answer-

ing. On the three test sets, our approach improves task accuracy metrics by 18.5% and interactivity by 46.3% on average compared to our best baselines, according to LLM judges. Beyond the tasks that the COLLABLLMs are fine-tuned on, we show COLLABLLMs are highly generalizable to other data domains.

Moreover, we perform a **large-scale and real-world user study** with 201 Amazon Mechanical Turkers (MTurkers), who are asked to write documents with the help of anonymous AI assistants, either COLLABLLM or non-collaboratively trained LLMs. COLLABLLM achieves impressive improvement with 17.6% increase in user satisfaction and yield user time savings of 10.4% on average. The qualitative analysis from MTurkers confirms our observations: non-collaboratively-trained LLMs passively agree with users, while COLLABLLM actively provide insightful questions and suggestions to guide writing processes.

## 2. Problem Formulation

In contrast to many existing tasks that are single-turn and require no human involvement beyond the initial query, our problem formulation reflects a real-world setting in which a user's underlying (implicit) goal is defined as $g$ in a multiturn conversational task. The conversation unfolds over multiple turns $t_j = \{u_j, m_j\}$, where $u_j$ is the user input and $m_j$ is the model's response at each turn $j = 1, \ldots, K$, where $K$ is the number of turns in the conversation.

At the $j$-th turn, the model generates its response based on the previous conversation turns $t_{1:j-1} = \{t_1, \ldots, t_{j-1}\}$ and the current user response $u_j$. For simplicity, we define historical conversation at $j$-th turn as $t_j^h = t_{1:j-1} \cup \{u_j\}$,

therefore, $m_j = M(t_j^h)$. The objective is to generate a sequence of model responses $\{m_j\}_{j=1}^K$ that effectively and efficiently achieve for goal $g$, *e.g.,* answering a math question, where goal achievement is assessed based on user satisfaction or an external evaluation function, such as accuracy by LLM judge. Formally, we define the objective as $R^*(t_{1:K} \mid g)$, where $R^*$ incorporate the achievement of task success and user experience factors such as time cost.

# 3. Unified Collaborative LLM Training

**Key Motivations**. Established LLM training frameworks, such as Reinforcement Learning from Human Feedback (RLHF) (Ouyang et al., 2022), focus on maximizing immediate rewards for single-turn tasks. This cause a misalignment between their single-turn objective and real-world multiturn objective $R^*(t_{1:K} \mid g)$. Precisely, the model's accumulative single-turn reward $\sum_{j=1}^{j=K} R(m_j \mid t_j^h)$ may not imply a higher final reward $R^*(t_{1:K} \mid g)$. *In fact, achieving high single-turn rewards at each turn may not imply a higher final reward*. For example, consider a task where the user's goal $g$ is to write an engaging article. A model trained with traditional RLHF might generate isolated responses, like drafting an introduction or listing conclusions. While these responses are helpful in isolation, they fail to consider how the sections flow together, resulting in an article that might not be cohesive and aligned with the user's goal.

*Instead, effective multiturn collaboration requires model responses that optimally contribute to the final reward*. The model should aim to align its responses with the user's goal $g$ by considering their impact on the entire conversation trajectory $t_{1:K}$. In the previous example, instead of generating a conclusion, asking, "*Should I maintain an engaging tone in the conclusion like the introduction?*" offers better long-term alignment with the goal.

## 3.1. Multiturn-aware Rewards

In Figure 1, our key insight is that effective multiturn collaboration relies on **forward-looking strategies**. Given a context ①, the model should consider how its response ② influences the subsequent turns of the conversation. To capture this, we design a ③ collaborative simulation module to estimate this impact. By ④ fine-tuning to distinguish between potential future conversations resulting from different responses, the model generates responses that align better with the overarching goal $g$.

This high-level design naturally aligns with causal effect estimation (Pearl, 2009; Pearl et al., 2016), which evaluates the interventional effects of an action in sequential decision-making. Appendix A provides further discussion on the connection between causal effect estimation and our

approach. More specifically, we define the Multiturn-aware Reward:

**Multiturn-aware Reward (MR):** *The multiturn-aware reward for model response $m_j$ at the $j$-th turn is given by:*

$$
\begin{aligned}
&\mathrm{MR}(m_j \mid t_j^h, g) \\
&= \mathbb{E}_{t_j^f \sim P(t_{j+1:K} \mid t_j^h \cup \{m_j\})} R^*(t_j^h \cup \{m_j\} \cup t_j^f \mid g) \quad (1) \\
&= \mathbb{E}_{t_j^f \sim P(t_j^f \mid t_{1:j})} R^*(t_{1:j} \cup t_j^f \mid g),
\end{aligned}
$$

*where $t_{1:j}$ denotes the conversation history up to and including the $j$-th turn, and $t_j^f = t_{j+1:K}$ represents the forward trajectory of turns following the $j$-th turn. The distribution $P(t_j^f \mid t_{1:j})$ models the possible forward conversations conditioned on the prior conversation history.*

However, computing Equation 1 remains challenging as it requires the following components: **(a) A conversation-level reward function**, $R^*(t \mid g)$, for evaluating an arbitrary multiturn conversation $t$, and **(b) a sampling strategy for obtaining forward conversations** $P(t_j^f \mid t_{1:j})$, which represents the forward conversation distribution. We elaborate on the two components in Section 3.1.1 and 3.1.2.

### 3.1.1. CONVERSATION-LEVEL REWARD FUNCTION

We approximate the conversation-level reward $R^*(t \mid g)$ with a combination of extrinsic (goal-specific) and intrinsic (goal-agnostic) metrics:

$$
R^*(t \mid g) \simeq R_{\text{ext}}(t, g) + R_{\text{int}}(t), \quad (2)
$$

where $R_{\text{ext}}(t, g)$ focuses on task success, and $R_{\text{int}}(t)$ evaluates user experience including efficiency and engagement.

- **Extrinsic Reward** $R_{\text{ext}}(t, g)$ measures how well the conversation achieves the user's goal $g$. Formally:

$$
R_{\text{ext}}(t, g) = S(\text{Extract}(t), y_g), \quad (3)
$$

where $\text{Extract}(t)$ extracts the final solution or response from the conversation $t$, especially for tasks requiring revisions or multi-step answers. $y_g$ is the reference solution for the goal $g$, *e.g.,* the ground truth solution for a math problem. And $S(\cdot, \cdot)$ evaluates task-specific metrics like accuracy or similarity. This ensures the conversation contributes directly to achieving the desired goal.

- **Intrinsic Reward** $R_{\text{int}}(t)$ prioritizes conversations that enhance user experience, defined as:

$$
R_{\text{int}}(t) = -\min[\lambda \cdot \text{TokenCount}(t), 1] + R_{\text{LLM}}(t), \quad (4)
$$

where we encourage conversational efficiency by penalizing excessive tokens that users read and write, with $\lambda$ controlling the penalty severity. This efficiency measure is bounded by 1 to maintain balance with other metrics.

The second term, $R_{\text{LLM}}(t)$, is assigned by an LLM-based judge (Zheng et al., 2023) on a 0–1 scale, evaluating user-valued objectives such as engagement / interactivity. Notably, additional conversational aspects, such as clarity, can be further integrated into the objective.

The conversation-level reward incorporates task-specific and human-centered metrics, encouraging the model to balance goal achievement, efficiency, and engagement.

### 3.1.2. FORWARD SAMPLING

To compute Eq. 1, we require samples from $P(t_j^f \mid t_{1:j})$, the distribution of forward conversation conditioned on the conversation history. A simple approach is to use Monte Carlo sampling, where the conversation is extended turn-by-turn until it concludes. However, this can be computationally expensive for computing reward for every model response. For a scalable approximation, we introduce a window size $w$ as a hyperparameter to limit the maximum number of forward turns considered in $t_j^f$. This reduces the computational cost while maintaining sufficient context.

More importantly, while real-world conversations could be gathered from human participants, sampling multiple forward conversations during training is costly and impractical. To further reduce cost and ensure scalability, we introduce a user simulator $U$.

**User Simulator:**. *A user simulator $U : \mathcal{T} \to \mathcal{U}$ is a function that maps a given conversation history $t \in \mathcal{T}$ to a user response $u \in \mathcal{U}$. Specifically, $U$ generates a probabilistic distribution $P(u \mid t)$ over possible user responses conditioned on the conversation history $t$, simulating realistic user behavior.*

Specifically, we prompt an LLM to role-play as users, explicitly asking the LLM to follow the same language style as the previous user turns, and injecting typical user behaviors. The user simulator operates with an implicit goal $g$, which it seeks to achieve over the course of the conversation. This design emulates real-world scenarios where users may have evolving needs, limited background knowledge, or require clarification, resulting in naturally unfolding multiturn conversations (Park et al., 2024).

### 3.2. Optimization & Synthetic Datasets

With the conversation-level reward function and forward sampling strategy, we can compute MR for any model response without requiring an additional reward model, which is often costly and slow to train. Unlike traditional single-turn reward approaches, MR explicitly accounts for the impact of a response on future conversations, promoting long-term collaboration.

Further, we employ reinforcement learning (RL) methods

such as PPO (Schulman et al., 2017) and DPO (Rafailov et al., 2023) to guide the model in navigating complex conversations. By optimizing for higher MR, the model learns to generate responses that enhance overall effectiveness and efficiency by the end of the conversation.

Moreover, MR can generate **high-quality synthetic conversations** (*cf.* Figure 8 in Appendix B) for both supervised fine-tuning (SFT) and DPO. For SFT, it iteratively selects top-ranked responses to build realistic, goal-directed conversation histories. For DPO, it constructs pairwise comparisons by ranking responses at each turn, distinguishing "chosen" and "rejected" pairs based on MR scores. The generated synthetic data aligns with multiturn objectives.

Overall, COLLABLLM enables scalable dataset generation and online RL training without human annotation, making it generalizable across diverse tasks. In Appendix 7, we compare COLLABLLM with related prompting- and training-based approaches, highlighting its contributions.

## 4. Experimental Setup[*]

For fine-tuning and evaluation, we create three multiturn datasets using publicly available data across diverse domains (Hendrycks et al., 2021; Zhuo et al., 2024; Chiusano, 2024): collaborative document editing, coding problem assistance, and multiturn mathematics problem solving.

To build a multiturn environment (Figure 3), we employ GPT-4o-mini as a user simulator LLM to role-play realistic user behaviors, given the target problem and conversation history. Our simulation-based evaluations are designed to closely mimic real-world interactions (Park et al., 2024). Unlike traditional single-turn tasks, our setup requires dynamic interactions over multiple turns to achieving a goal. The three interactive datasets are:

`MediumDocEdit-Chat`: Document editing requires iterative feedback and refinements across multiple turns to ensure coherence and alignment with user intent. We sample 100 Medium articles as goal documents, which are summarized into target problems to guide the user simulator. After each interaction, task performance is evaluated using the **BLEU** score, measuring similarity between the extracted document and the original articles.

`BigCodeBench-Chat`: Coding tasks inherently require multiturn interactions, such as clarifying requirements and debugging. We sample 600 coding problems from BigCodeBench (Zhuo et al., 2024) as the target problems given to the user simulator. For evaluation, we compute the average **Pass Rate (PR)** of code at the end of the interactions.

---

[*]Dataset and training details in Appendix B; all prompts (*e.g.,* prompts of user simulator and LLM judges) in Appendix D.

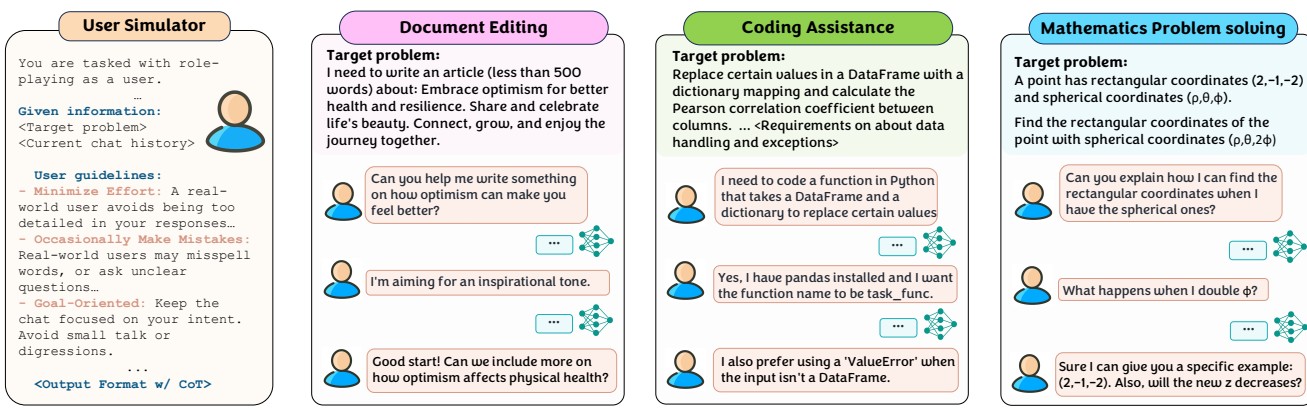

Figure 3: Simulated Multiturn Environment for Evaluation. Our evaluation pipeline simulates real-world collaborations by prompting an user simulator LLM to emulate diverse behaviors and personalities in multiturn conversations.

**MATH-Chat**: Math problem solving often requires addressing implicit assumptions, verifying intermediate steps, and clarifying reasoning. We sample 200 level-5 math problems from MATH (Hendrycks et al., 2021) to prompt the user simulator, which interacts with the LLMs. Task success is measured by the **accuracy (ACC)** of the final solution, as evaluated by an LLM judge.

In addition to the above task-specific metrics, we incorporate two task-agnostic scores across all datasets: **1) Average Token Count**, which quantifies the average number of tokens generated by the LLM per conversation, reflecting interaction efficiency. **2) Interactivity (ITR)**, which evaluates engagement levels using an LLM judge (Claude-3.5-Sonnet), with scores rescaled to an upper bound of 1.

**Fine-tuning COLLABLLMs.** COLLABLLMs are based on Llama-3.1-8B (Llama Team, 2024) with LoRA fine-tuning (Hu et al., 2022). We train four model variants: **1) Offline models**: SFT and Offline DPO are fine-tuned on pre-generated multiturn conversational datasets guided by Multiturn-aware Rewards (MR) (*cf.* Section 3.2). **2) Online models**: PPO and Online DPO are further trained from the SFT and Offline DPO models, respectively. The model during online fine-tuning is involved in the collaborative simulation to compute MRs, which, in turn, dynamically adjust the model preference.

**Baselines**. We compare COLLABLLMs against (1) the pretrained Llama-3.1-8B (*Base*), (2) the base model with proactive prompt engineering (*Proactive Base*), which encourages follow-up and clarification questions.

## 5. Results of Simulated Experiments

We present the results in Table 1 and the takeaways are:

**Prompt engineering is helpful, but limited in terms of performance gains and flexibility**. Proactive Base improves base model performance by encouraging follow-up questions and clarifications. For example, it increases BLEU on MediumDocEdit-Chat from 32.2% to 35.0% and reduces read tokens by 0.31k compared to the base model. However, these gains are modest and do not fully address the challenges of multiturn collaboration. We observe that prompting strategies remain rigid, relying on predefined instructions rather than adapting dynamically to user needs. For instance, the model sometimes asks clarification questions even when unnecessary, leading to redundant interactions that disrupt conversation flow.

**COLLABLLM improves task performance, efficiency, and engagement**. COLLABLLM achieves 18.5% superior task-specific performance, 13.3% more efficient conversations, and 46.3% enhanced interactivity compared to the best baselines. We highlight that COLLABLLM engage in more meaningful collaborations, with ITR shows substantial gains. For MediumDocEdit-Chat, the Online DPO model increases ITR from 0.46 to 0.92. Moreover, our framework significantly improves conversational efficiency by minimizing the content users need to review to arrive at the final solution. For MATH-Chat, Online DPO decreases token count per conversation by 1.03k compared to the base model.

### 5.1. Ablations on Reward Mechanisms (Figure 9)

To investigate how components contribute to COLLABLLM's superior performance, we conduct an ablation study focusing on the reward mechanisms used during fine-tuning. We evaluate the following reward mechanisms:

- **Variants of Multiturn-aware Reward**: We vary the forward sampling window size $w = 1, 2, 3$ to assess their ability to capture long-term conversational effects through simulated collaborations.
- **Immediate Rewards** evaluate the model's immediate response based on: *1) Helpfulness*: Assessed by an LLM judge; *2) Extrinsic Reward*: Focuses on task-specific

Table 1: Evaluation results on our multiturn datasets. Green zone: Baselines; Orange zone: Variants of COLLABLLMS. *Rel. Improv.* indicates the relative improvements of CollabLLMs trained with Online DPO over Proactive Base.

| | MediumDocEdit-Chat | | | BigCodeBench-Chat | | | MATH-Chat | | |
|---|---|---|---|---|---|---|---|---|---|
| | BLEU ↑ | #Tokens$(k)$ ↓ | ITR ↑ | PR ↑ | #Tokens$(k)$ ↓ | ITR ↑ | ACC ↑ | #Tokens$(k)$ ↓ | ITR ↑ |
| Base | 32.2 | 2.49 | 46.0 | 9.3 | 1.59 | 22.0 | 11.0 | 3.40 | 44.0 |
| Proactive Base | 35.0 | 2.18 | 62.0 | 11.0 | 1.51 | 33.7 | 12.5 | 2.90 | 46.0 |
| SFT | 35.2 | 2.21 | 68.0 | 11.7 | 1.35 | 42.0 | 13.5 | 2.88 | 58.0 |
| PPO | 38.5 | 2.00 | 78.0 | 14.0 | 1.35 | 40.7 | 13.0 | 2.59 | 52.0 |
| Offline DPO | 36.4 | 2.15 | 82.0 | 12.3 | 1.35 | 46.7 | 15.5 | 2.40 | 50.0 |
| Online DPO | 36.8 | 2.00 | 92.0 | 13.0 | 1.31 | 52.0 | 16.5 | 2.37 | 60.0 |
| Rel. Improv. | 5.14% | 8.25% | 48.3% | 18.2% | 13.2% | 54.3% | 32.0% | 18.3% | 36.4% |

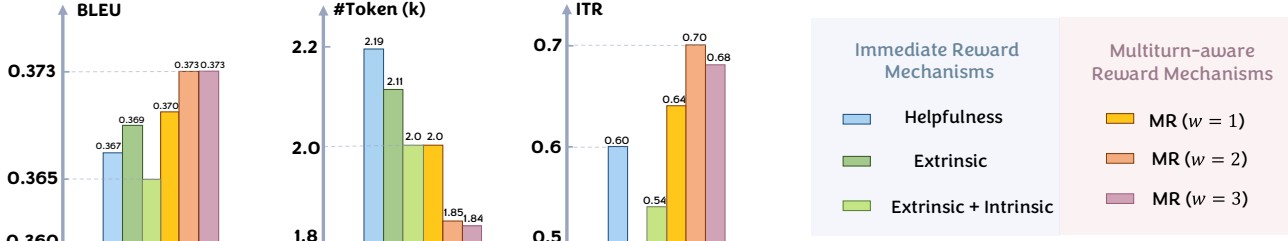

Figure 4: Selected Ablation Study of Reward Mechanisms on `MediumDocEdit-Chat`. This figure compares three immediate reward mechanisms with three MR variants. MR consistently improves task-specific performance (BLEU), conversational efficiency (# Tokens), and interactivity (ITR). See Appendix B.3 for the full results.

metrics like BLEU while ignoring intrinsic factors such as efficiency; *3) Extrinsic + Intrinsic Reward*: Combines task-specific metrics with efficiency and interactivity measures. This can be seen as a special case of the multiturn-aware reward function with $w = 0$.

We present results in Figure 9. Interestingly, expanding the forward sampling window $w$ within the range generally enhances performance and efficiency by better capturing future interactions. Notably, MR with $w = 2$ balances the gains and additional costs to conduct forward sampling, making it well-suited for large-scale fine-tuning. In contrast, immediate rewards, even with extrinsic and intrinsic components, fall short as they ignore long-term impact. These findings validate the positive impact of the forward sampling strategy in MRs.

### 5.2. Case Study (Figure 5 & 6)

We now offer a deeper insight into COLLABLLM's behavior as shown in Figure 5. In this example, the user request to tokenize a text file is inherently open-ended due to unspecified factors, such as the NLTK environment, tokenizer selection, and optional preprocessing steps. The base LLM makes several arbitrary assumptions, applying lowercase conversion and stopword removal without user confirmation. The user simulator later corrects these assumptions, but the final solution remains incorrect due to missing stopwords. In contrast, COLLABLLM actively clarifies user intent by seeking confirmation on key decisions, ensuring an aligned final solution with a 100% Pass Rate. This approach also reduces user effort with lower token usage.

In Figure 6, we compare different reward mechanisms for responses A and B of Figure 5, to confirm that these rewards work as intended. The helpfulness rewards favor response A due to its seemingly more well-round output. Extrinsic rewards assign zero scores to both, as A provides an incorrect solution and B defers answering. Extrinsic + Intrinsic rewards slightly favor B for efficiency and engagement. Interestingly, MR assigns significantly higher rewards to B, especially at $w = 2$ and $w = 3$, since the response obtains useful information and provide a precise answer within the future interaction window.

### 5.3. Model Generalization (Table 2)

Modern foundation models are expected to generalize across a diverse range of tasks beyond their training domain. A key question is whether collaborative behaviors learned by COLLABLLM during fine-tuning transfer effectively to new tasks without additional adaptation.

We assess COLLABLLM, trained with online DPO on `BigCodeBench-Chat` (the coding assistance task), on Abg-CoQA (Guo et al., 2021), a question-answering (QA) benchmark where questions are labeled as ambiguous or non-ambiguous (*cf.* Appendix E). We categorize the model's responses into two actions—asking a clarifying question or providing a direct answer—and evaluate action-level accuracy within each question type. As shown in Table 2, GPT-4o and Llama-3.1-8B rarely ask clarifying questions regardless of ambiguity. In contrast, COLLABLLM

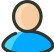

Hi, I need help writing a function that tokenizes a text file using NLTK.
It should be named `task_func` and take a `file_path` as a parameter. Could you help me with that?

**Non-collaborative LLM**

Before we start, make sure you have the NLTK installed.

You can install it using pip:
```
pip install nltk
```

Here's the function that tokenizes a text file:
```python
import nltk
from nltk.tokenize import word_tokenize
from nltk.corpus import stopwords
import string

def task_func(file_path):

    # Initialize the NLTK data needed for processing text
    nltk.download('punkt')
    nltk.download('stopwords')

    # Open the file in read mode
    with open(file_path, 'r') as file:
        # Read the content of the file
        text = file.read()

    # Convert the text to lowercase
    text = text.lower()

    # Remove punctuation
    text = text.translate(str.maketrans('', '', string.punctuation))

    # Tokenize the text
    tokens = word_tokenize(text)

    # Remove stopwords
    stop_words = set(stopwords.words('english'))
    tokens = [token for token in tokens if token not in stop_words]

    return tokens
```
This is a basic example of tokenization. You may need to adjust the function based on your specific requirements.

**Ours: CollabLLM**

I'd be happy to help with your function! Could you provide the following:

1. The NLTK library's version you're using (or the version you've installed).
2. The specific NLTK tokenizer you'd like to use (e.g., WordTokenizer, RegexpTokenizer, etc.).
3. Confirm if you want me to the function to handle any potential errors (e.g., file not found, incorrect file path, etc.)

Also, I assume I don't need remove any punctuations in the text. Please let me know otherwise.

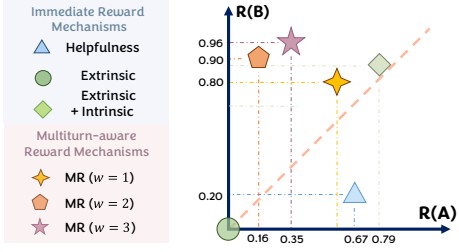

I haven't noticed what version of the NLTK I have, but I installed it recently if that helps. I would like to use the basic word tokenization. The error handling part doesn't have to be extensive, maybe just something simple if you have suggestions. Finally, you can keep the punctuations as they are.

Here's a simple implementation of the `task_func` function that uses NLTK's basic word tokenization:
```python
import nltk
from nltk.tokenize import word_tokenize

def task_func(file_path):
    try:
        with open(file_path, 'r') as file:
            text = file.read()
            tokens = word_tokenize(text)
            return tokens
    except FileNotFoundError:
        print(f"Error: The file '{file_path}' was not found.")
        return None
```
If you're using NLTK for the first time, you might need to run `nltk.download('punkt')`. Do you need further assistance on that?

Figure 5: Case study on `BigCodeBench-Chat`. The non-collaborative LLM assumes user needs, adding unnecessary steps like punctuation and stopword removal. In contrast, COLLABLLM clarifies tokenizer preferences, error handling, and package installation, leading to a solution that precisely aligns with user intent.

Figure 6: Reward comparison for response A and B of Figure 5 shows different preferences.

| | Action-level Accuracy | | Macro Metric | |
| | Ambiguous | Non-Ambiguous | Accuracy | F1 |
|---|---|---|---|---|
| GPT-4o | 15.44% | 95.60% | 55.52% | 56.62% |
| Base (Llama-3.1-8B) | 16.26% | 90.40% | 53.33% | 53.31% |
| COLLABLLM | 52.84% | 72.32% | 62.58% | 55.08% |

Table 2: Zero-shot generalization to `Abg-CoQA`, a conversational QA benchmark to identify ambiguity. We assess action-level accuracy, measuring whether the model asks a question for ambiguous inputs and provides a direct answer for non-ambiguous ones.

proactively asks questions about 50% of the time while maintaining high accuracy on unambiguous inputs. This behavior leads to the highest Macro Accuracy across both ambiguous and non-ambiguous sets and improves Macro F1 over the base model, while leaving room for further improvement against GPT-4o. These results suggest that **COLLABLLM effectively generalizes its learned collaborative strategies beyond its training domain**.

## 6. Real-world User Study

**Setup**. We conduct a large-scale user study using Amazon Mechanical Turk with 201 participants. Each participant is assigned a document type—randomly selected to be either blog post, creative writing, or personal statement—and chooses a topic from a predefined set. To simulate real-world scenarios where users have only a rough idea of the task, they are first asked to provide brief responses to topic-related questions. Participants then engage in at least eight turns of conversation with an anonymized AI assistant, which can be Base, Proactive Base, or COLLABLLM. Every three turns, they provide an interaction rating based on their experience so far. After the conversation, participants rate the final document quality and overall interaction. All ratings are in a scale from 1 to 10. We also record the total interaction duration to assess efficiency. The detailed user study setup is provided in Appendix F.

**Quantitative Results (Figure 7)**. Across multiple metrics, COLLABLLM consistently outperforms the baselines. It achieves an average document quality score of 8.50. Specifically, 91.4% of participants rate COLLABLLM's **document quality** as "good" (score 8–9), and 56.9% as "very good" (score 9–10), compared to 88.5% and 39.3%

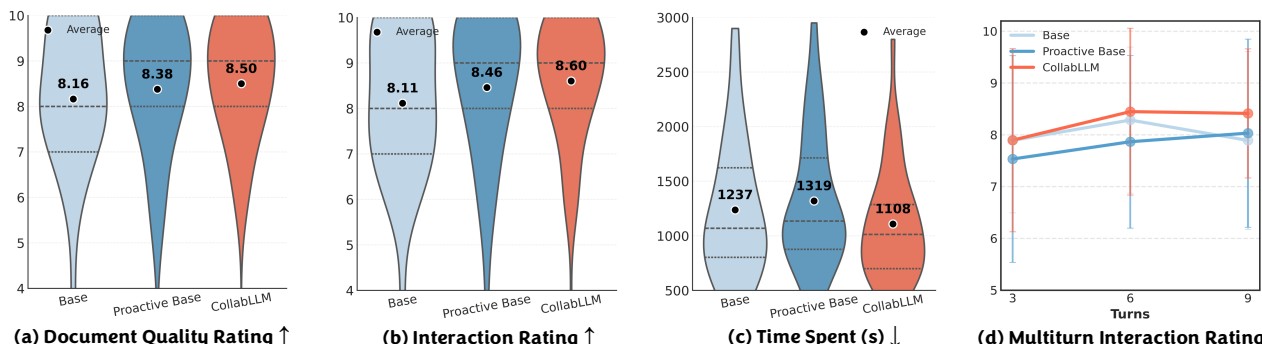

(a) Document Quality Rating ↑  (b) Interaction Rating ↑  (c) Time Spent (s) ↓  (d) Multiturn Interaction Rating ↑

Figure 7: Our real-world user study includes 201 participants interacting with an anonymized AI assistant randomly sampled from Base, Proactive Base, and COLLABLLM. Participants rate (a) document quality and (b) overall interaction experience, with additional assessments (d) every three turns. We also measure (c) user spent time to evaluate efficiency.

Table 3: Representative Feedback from Human Participants.

| Model | Strengths | Weaknesses |
|---|---|---|
| Base | *"Follows great instruction and does exactly what I'm asking it to do.", "It can create a nice form of an outline to work with."* | *"The AI just agreed with me on pretty much everything. There was no discussion", "I didn't really like that it kept coming up with different options"* |
| Proactive Base | *"It is very organized and it actually asks you for feedback after writing the revision."* | *"The AI seemed to be very redundant and asked me the same questions over and over."* |
| COLLAB LLM | *"Asking questions and making you think of things you never thought of", "The AI really helped me with focusing on one part of the story at a time.", "It helped really well to navigate what to say and what information is needed"* | *"The AI assistant was not up to date enough to help with this recent sporting event. The AI assistant also asked me to repeat information I had already given it."* |

for Base (Llama-3.1-8B), respectively. Similarly, 63.8% of participants find COLLABLLM **highly engaging**, while only 42.6% report the same for Llama-3.1-8B.

Interestingly, for **multiturn interaction**, the Base model shows a declining trend in ratings from turns 6–9, indicating reduced user experience in longer conversations. In contrast, both COLLABLLM and Proactive Base exhibit increasing ratings over time, with COLLABLLM consistently achieving higher average ratings every three turns compared to Proactive Base. This suggests that COLLABLLM maintains sustained engagement more effectively.

Moreover, COLLABLLM improves task efficiency, reducing **time spent** by 10.4% compared to the Base model and by 15.6% relative to Proactive Base. While Proactive Base is prompted to maintain conciseness, it frequently asks unnecessary questions, causing lower efficiency. In contrast, COLLABLLM strikes a more streamlined user experience.

**Qualitative Results (Table 3)**. We collected a total of 180 strengths and 180 weaknesses across the three models. Table 3 presents representative feedback, while we summarize here the mddels' strengths and weaknesses: The base model generates coherent content while effectively follow user instructions, but it sometimes struggles with maintaining context in long texts, and can be overly verbose or repetitive in its responses. Proactive Base excels in responsiveness and adapting to user input but struggles with memory retention, and could produce repetitive or overly structured content. On the other hand, COLLABLLM is highly engaging, effectively guiding users through writing, adapting seamlessly to feedback. However, users also point out that COLLABLLM can occasionally feel bland, lack of up to date information, and require additional effort to personalize the output. Overall, COLLABLLM enhances collaboration by guiding users through an interactive and iterative refinement process, yet future improvements should focus on increasing personalization, creativity, and real-time knowledge integration to further optimize human-LLM collaboration.

## 7. Related Work

**Non-collaborative LLM training**. Existing LLM training frameworks, including pre-training, supervised fine-tuning (SFT), and reinforcement learning (RL) (Rafailov et al., 2023; Schulman et al., 2017; Ouyang et al., 2022; Lee et al., 2024), primarily optimize for next-turn response quality. Standard RL methods such as Proximal Policy Optimization (PPO) (Schulman et al., 2017) apply rewards to individual model responses without accounting for their long-term impact on conversation trajectories. While effective

Table 4: Compare CollabLLM with Selected Works. (1) Task-Agnostic, assessing whether the approach applies across diverse domains rather than being task-specific; (2) Versatile Interaction, evaluating its ability to support diverse strategies for intent discovery and efficient task completion beyond predefined behaviors; (3) User-Centric, determining whether engagement, efficiency, and intent discovery are explicitly considered; and (4) Causal & Objective-Aligned Reward, measuring whether reward estimation captures causal effects on future interactions and optimizes for long-term task success.

| | Task-Agnostic | Versatile Interaction | User-Centric | Causal & Objective-Aligned Reward |
|---|---|---|---|---|
| ClarifyGPT (Mu et al., 2023) | ✗ | ✗ | ✗ | - |
| STaR-GATE (Andukuri et al., 2024) | ✔ | ✗ | ✗ | - |
| MTPO (Shani et al., 2024) | ✔ | ✔ | ✗ | ✗ |
| CollabLLM | ✔ | ✔ | ✔ | ✔ |

for single-turn objectives, these approaches fail to capture how responses influence user intent discovery and long-term task success (Amershi et al., 2019; Zamfirescu-Pereira et al., 2023; Wang et al., 2024; Kim et al., 2024).

**Prompting techniques for multiturn interaction**. Prior work has explored prompting strategies to enhance LLM interactivity, particularly for clarification questions (Keh et al., 2024; Mu et al., 2023; Zhang & Choi, 2023; Chi et al., 2024; Kim et al., 2023; Deng et al., 2023b; Zhao & Dou, 2024) and mixed-initiative dialogues (Deng et al., 2023a; Chen et al., 2023; Liao et al., 2023). For instance, Mu et al. (2023) prompt LLMs to ask clarification questions when code generation requests are ambiguous. However, such prompting-based approaches are constrained by predefined interaction patterns, limiting adaptability across different tasks and conversation stages. Moreover, their reliance on fixed prompts reduces generalization, as demonstrated in our experiments where proactive prompting fails to match the effectiveness of our fine-tuned models.

**Learning-based methods for multiturn interaction**.

- **LLMs for generating clarification questions:** Beyond prompting, prior studies have explored supervised fine-tuning (Andukuri et al., 2024), RL fine-tuning (Chen et al., 2024; Zamani et al., 2020; Erbacher & Soulier, 2023), and active learning (Pang et al., 2024) to train models to ask clarification questions. For example, Chen et al. (2024) use Direct Preference Optimization (DPO) to encourage models to request clarifications. However, like prompting approaches, these methods primarily focus on clarification questions and do not generalize to broader multiturn collaboration strategies.

- **Multiturn training for LLMs:** Recent benchmarks (Abdulhai et al., 2023; Kwan et al., 2024) evaluate LLMs' performance in multiturn settings, measuring the goal orientation and planning capabilities of models across interactions. Several studies extend RLHF to multiturn settings by optimizing trajectory-level rewards (Shani et al., 2024; Zhou et al., 2024; Gao et al., 2024; Shi et al., 2024b; Zhang et al., 2025). Other works (Xu et al., 2023; Deng et al., 2024) leverage self-

chat or self-play to enhance model adaptation. However, these methods primarily rely on post-hoc trajectory-level data, learning from observed conversations rather than explicitly modeling the causal effect of individual responses on task success (see Appendix A for further explanations). Additionally, they often overlook open-ended tasks such as document generation (Faltings et al., 2023; Jiang et al., 2024), where user responses can be highly diverse, and users may have limited capacity to read and refine lengthy model outputs.

**User simulators for enhancing AI systems**. Recent works employ user simulators to enhance dialogue systems (Shi et al., 2019; Tseng et al., 2021) and LLMs (Hong et al., 2023; Hu et al., 2023; Faltings et al., 2023). Recently, Hong et al. (2023) leverage LLMs to create diverse synthetic dialogues with varying user personas to train smaller dialogue models. CollabLLM differs in leveraging user simulators in forward sampling to account for long-term effect in both offline and online training.

In Table 4, we compare CollabLLM with related methods across four key dimensions. CollabLLM is a general, user-centric, and multiturn-aware framework that leverages more accurate reward estimation to better align with real-world objectives, enhancing user satisfaction and streamlining human-LLM interactions.

# 8. Conclusion

Multiturn human-LLM collaborations are increasingly prevalent in real-world applications. Foundation models should act as collaborators rather than passive responders, actively uncovering user intents in open-ended and complex tasks—an area where current LLMs fall short. The key insight of CollabLLM is making LLMs more multiturn-aware by using forward sampling to estimate the long-term impact of responses. Through extensive simulated and real-world evaluations, we demonstrate that CollabLLM is highly effective, efficient, and engaging, while also generalizing well to new tasks and interactions, advancing the frontiers of human-centered LLMs.

## Acknowledgments

We thank Doug Burger, Vishal Chowder, Jeevana Priya Inala, Hoifung Poon, Swadheen Shukla, Chandan Singh, Desney Tan and Chenglong Wang, as well as members of the Deep Learning and Health Futures groups at Microsoft Research for helpful discussions. We thank lab members in Leskovec and Zou's labs for discussions and for providing feedback. We also gratefully acknowledge the support of NSF under Nos. OAC-1835598 (CINES), CCF-1918940 (Expeditions), DMS-2327709 (IHBEM), IIS-2403318 (III); Stanford Data Applications Initiative, Wu Tsai Neurosciences Institute, Stanford Institute for Human-Centered AI, Chan Zuckerberg Initiative, Amazon, Genentech, GSK, Hitachi, SAP, and UCB.

## Impact Statement

This paper presents work aimed at making AI more user- and **human-centric**, which, in our view, yields a positive societal impact. Most current work on AI and its evaluation focuses on fully automated tasks, with no user involvement in solving the task or optimization for a collaborative experience with users. This has serious societal drawbacks, given issues such as AI hallucinations (Huang et al., 2025), biases (Gallegos et al., 2024), and unsafe language (Shi et al., 2024a) that arise from a lack of human oversight. The common focus on having AI models autonomously complete tasks also ignores the reality that many scenarios have humans present regardless of the level of automation, and that not priming AI models to proactively seek human help, feedback, or clarifications misses an opportunity to make generative AI more accurate, effective, and safe. This consideration would also help increase the adoption of AI in safety-critical scenarios, such as medical decision-making tasks (Liu et al., 2024), in which we believe AI models should be inclined to seek confirmation or verification (Gero et al., 2023) from an expert in case of uncertainty—a behavior that is mostly absent in current state-of-the-art LLMs.

Since the models in this work are trained collaboratively and aim to better align with user intent, concerns may arise regarding users with malevolent goals. However, we argue that COLLABLLM can help **mitigate safety risks** in such cases—at least when used with LLMs that have been aligned for safety (as is the case for all models used in this work). Safety-aligned LLMs generally refuse to respond to unsafe queries, which often leads malicious users to obscure their true intentions in order to bypass safeguards. This is where our approach offers an advantage: COLLABLLM often seeks to clarify user intent, creating additional opportunities to detect misuse. For example, malicious users might unintentionally reveal their actual goals, or their vagueness and refusal to disclose motivations could raise red flags—potentially providing the LLM with further cues for identifying unsafe behavior. As presented in Appendix C, we conducted various safety experiments and show that COLLABLLM performs no worse than an equivalent non-collaboratively trained model in terms of safety.

The data collected in our study involves human participants recruited through Mechanical Turk. We took several measures to ensure the privacy of these workers in the document creation tasks. First, we asked workers to confirm that they were willing to share the text they wrote as part of a public dataset. Second, we urged them not to include any personally identifiable information (PII) in their writings and to focus only on topics of public knowledge or fictitious stories. Third, we scanned the collected data to ensure that no PII was included. For the final version of the dataset, we will recruit additional workers to manually review each collected conversation to ensure that no PII or other safety issues (e.g., offensive language) exist in the data. Mechanical Turk workers were paid $10 per conversation. Given that conversations averaged 28.4 minutes, including break times, this means workers were paid more than $20 per hour on average—above the minimum wage in the country where the data was collected.

This work presents one of the first attempts to train LLMs in such human-centric environments. To promote future research in this societally beneficial direction, we release all the code, models, data, benchmarks, and user simulators described in this work.

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

# A. Supplementary Discussion

## A.1. Connection Between Multiturn-aware Reward and Causal Inference

Our approach naturally aligns with causal inference principles, as it aims to quantify how a model's response influences the future trajectory of a conversation. This aligns with the fundamental goal of **causal effect estimation**, which seeks to isolate the impact of an intervention—in this case, a model response—on long-term outcomes.

From a causal perspective, given a conversation history $t_j^h$ at turn $j$, the **causal effect** of a model response $m_j$ on the final conversation trajectory can be expressed using **front-door adjustment** (Pearl, 2009; Pearl et al., 2016):

$$\sum R^*(t_{1:K} \mid g)P(t_{1:K} \mid t_j^h)P(t_j^h) = \sum R^*(t_{1:K} \mid g)P(t_{1:K} \mid t_j^h) = \mathbb{E}_{t_{1:K} \sim P(t_{1:K}|t_j^h)} R^*(t_{1:K} \mid g). \tag{5}$$

This equation captures the expected long-term reward of a conversation conditioned on the model's response at turn $j$. It explicitly accounts for how $m_j$ intervenes in the conversation, influencing future turns and, ultimately, task success.

## A.2. Distinction from Other Multiturn Training Frameworks

Existing multiturn trajectory-based training frameworks (Shani et al., 2024; Zhou et al., 2024; Gao et al., 2024) primarily rely on learning from observed trajectory-level rewards. These methods estimate the utility of responses by assigning rewards post hoc to completed conversations, typically training models to prefer higher-rated conversations over lower-rated ones. However, this approach is fundamentally **observational**—it captures statistical associations between responses and final outcomes, without disentangling how individual responses causally influence future turns. For example, in MTPO (Shani et al., 2024), the learning signal remains coarse-grained: rewards are assigned at the trajectory level, and the influence of specific turns within a conversation remains confounded and indirect.

In contrast, our Multiturn-aware Reward (MR) framework **intervenes** on individual model responses and uses forward simulation to generate alternative future trajectories. This allows the model to estimate the **counterfactual impact** of different responses at each turn, thereby enabling fine-grained optimization. By leveraging causal effect estimation, MR training moves beyond passive imitation of high-reward conversations and instead actively selects responses to maximize long-term task success. This interventional approach provides turn-level credit assignment that is critical in dynamic human-LLM interactions, where user needs evolve and the consequences of early decisions compound over time.

# B. Experimental Details

## B.1. Dataset Generation for Offline Training

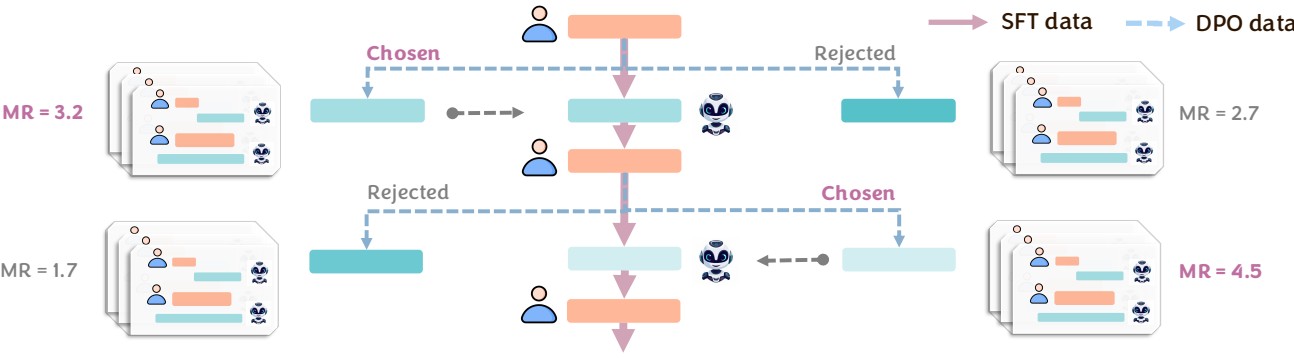

Figure 8: Generating high-quality conversation data with Multiturn-aware Rewards (MR).

The Multiturn-aware Reward (MR) function enables the generation of high-quality synthetic conversation datasets for training. Given a user query, multiple LLM responses are sampled and ranked based on their MR scores, with higher-ranked responses designated as *Chosen* and lower-ranked as *Rejected*. To simulate natural conversational flow, the first turn from the chosen response's forward interaction window is appended to the prompt for the next turn, iteratively extending the conversation until completion. Solid red arrows denote data collection for Supervised Fine-Tuning (SFT), while dashed blue arrows indicate preference data construction for Direct Preference Optimization (DPO). This approach systematically curates multiturn conversations that enhance both response quality and collaborative efficiency, both of which are explicitly captured by MR.

Given (1) a user simulator LLM, *e.g.,* GPT-4o-mini, (2) an assistant LLM, GPT-4o, and (3) arbitrary tasks with defined task-specific metric, we can simulated and generate high-quality conversations following Figure 8. We create the following training datasets in this simulated environments.

Table 5: Statistics of conversational datasets created from MR. Chosen/Rejected MR indicates the mean and standard deviation (mean $\pm$ std) of MRs for chosen and rejected responses (*cf.* Figure 8).

|  | # Train | # Turns | Average # Turns | Chosen MR | Rejected MR |
|---|---|---|---|---|---|
| MediumDocEdit-Chat | 500 | 2,303 | 4.61 | 0.312 $\pm$0.104 | 0.246 $\pm$0.113 |
| BigCodeBench-Chat | 500 | 2,627 | 5.25 | 0.494 $\pm$0.621 | 0.207 $\pm$0.763 |
| MATH-Chat | 500 | 2,527 | 5.05 | 0.863 $\pm$0.524 | 0.547 $\pm$0.502 |

## B.2. Training Details

**Hyperparameters (Table 6).** We provide the hyperparameters for COLLABLLM fine-tuning.

Notably, COLLABLLM relies on a minimal set of hyperparameters, using the same window size and sample size for computing MRs across multiple datasets. The penalty factor on token count, $\lambda$, is set lower for MediumDocEdit-Chat compared to BigCodeBench-Chat and MATH-Chat, as document lengths in MediumDocEdit-Chat can vary significantly and may be easily bounded by 1 in Eq. 4 if $\lambda$ is too large. **Training Cost (Table 7).** We compute average

Table 6: Hyperparameters for LoRA configuration, different stages of fine-tuning, and COLLABLLM-specific fine-tuning.

| LoRA Configuration | |
|---|---|
| Rank $r$ | 32 |
| Scaling factor $\alpha$ | 16 |
| Dropout | 0.1 |
| Bias | False |

| Fine-Tuning Hyperparameters | | | | |
|---|---|---|---|---|
|  | SFT | Offline DPO | Online DPO | PPO |
| Learning rate | 1e-5 | 5e-6 | 5e-6 | 2e-6 |
| Total batch size | 64 | 64 | 32 | 64 |
| Number of epochs | 3 | 8 | 1 | 5 |

| COLLABLLM-specific Hyperparameters | | | |
|---|---|---|---|
|  | MediumDocEdit-Chat | BigCodeBench-Chat | MATH-Chat |
| Window size $w$ | 2 | 2 | 2 |
| Sample size for MR | 3 | 3 | 3 |
| Penalty $\lambda$ | 1e-4 | 5e-4 | 5e-4 |

statistics over 100 future conversations on MediumDocEdit-Chat, the document editing task, which incurs the highest computational overhead among the three tasks. The table shows that even at the largest window size ($w = 3$), the total per-sample cost remains low, suggesting that our multi-turn training setup is financially practical. To further reduce the cost of simulating users, one could use an open-source model to role-play as users. **Unfortunately, at the current stage, we find that open-source models generally perform poorly, often getting "confused" and starting to solve problems as an assistant rather than acting as a user.** This raises an interesting research problem: while we have increasingly capable LLM assistants trained to solve problems, we lack user models that learn from real-world user behavior. Building better user models could be valuable for running simulations in real-world applications.

|  | Policy Model Input Tokens (k) | Policy Model Output Tokens (k) | Policy Model Time (s) | User Simulator Input Tokens (k) | User Simulator Output Tokens (k) | User Simulator Cost ($) |
|---|---|---|---|---|---|---|
| $w = 1$ | 0.89 | 0.42 | 7.41 | 1.85 | 0.26 | 0.00174 |
| $w = 2$ | 2.55 | 0.91 | 15.84 | 4.55 | 0.69 | 0.00439 |
| $w = 3$ | 4.13 | 1.22 | 21.72 | 7.18 | 1.06 | 0.00685 |

Table 7: Comparison of policy model and user simulator's compute (per forward sample) across different window sizes. We use GPT-4o-mini as the user simulator. The results are averaged over 100 forward sampled conversations.

## B.3. Full Ablation Results

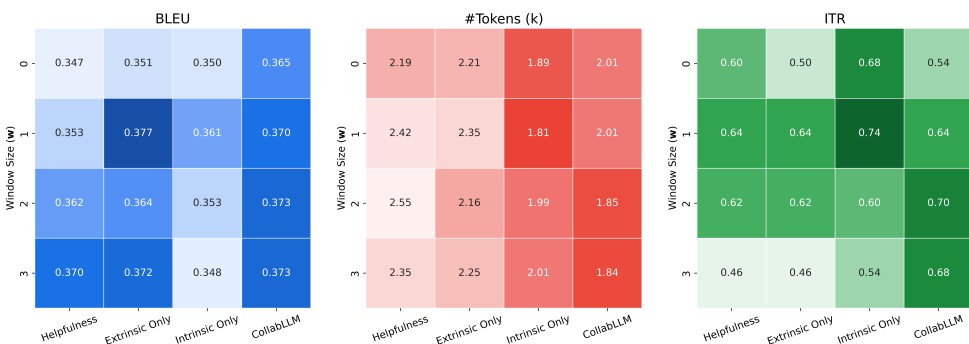

Figure 9: Full ablation study showing the impact of different reward types (Helpfulness, Extrinsic Only, Intrinsic Only) and window sizes ($w$) on BLEU, token count (in thousands), and Interactivity Rate (ITR). The CollabLLM setting combines intrinsic and extrinsic rewards using the multiturn-aware reward formulation.

To further understand the source of performance improvements, we conduct a full ablation by training models with isolated reward signals—*Helpfulness*, *Extrinsic Only*, and *Intrinsic Only*—across window sizes $w \in \{0, 1, 2, 3\}$. The resulting BLEU, token usage, and ITR scores are reported in Figure 9.

We make three key observations:

- **Helpfulness alone** leads to marginal improvements in BLEU and ITR, but significantly increases token usage, especially at larger window sizes, suggesting verbosity rather than improved efficiency or interactivity.
- **Extrinsic-only reward** achieves strong BLEU scores (e.g., 0.377 at $w = 1$), indicating good task alignment. However, it underperforms in ITR and often generates longer responses.
- **Intrinsic-only reward** improves ITR at $w = 1$ (e.g., 0.74), but offers lower BLEU and comparable or slightly lower token efficiency, indicating better interactivity at the expense of task success.

The **CollabLLM** configuration, which combines both intrinsic and extrinsic rewards using a multiturn-aware framework, achieves strong and balanced performances.

Note that the choice of reward type (intrinsic or extrinsic) is independent of the multiturn-aware reward design. In practice, one can flexibly plug in different reward signals, which are then used to evaluate the responses' long-term impact through forward sampling.

## C. Safety Evaluation

As the models in this work are collaboratively trained and designed to be more aligned with the user's intent, concerns may arise if a user happens to have malevolent intentions. However, we note that COLLABLLM models were finetuned from Llama-3.1-8B, which has been aligned for safety—so jailbreaking COLLABLLM still poses a significant challenge. To determine whether collaborative training weakens the safety features inherent to a model (Llama-3.1-8B) that has undergone significant alignment steps for safety, we performed an adversarial evaluation using the Azure AI Evaluation SDK[†] and prompted both the baseline and COLLABLLM with various offensive queries intended to elicit unsafe responses.

Specifically, we performed the following steps:

- **Adversarial query selection:** We used the SDK's `AdversarialSimulator` to generate adversarial queries (e.g., queries encouraging the LLM to produce hateful comments). We then used the SDK's harm evaluators (`ViolenceEvaluator`, `SexualEvaluator`, `SelfHarmEvaluator`, `HateUnfairnessEvaluator`) to categorize each query into one of four harm types: violence, sexual, self-harm, and hate. For each query, we used the highest score among the four evaluators to determine its harm category. We randomly selected 20 adversarial queries per harm category, resulting in a total of 80 queries.

[†] https://learn.microsoft.com/en-us/python/api/overview/azure/ai-evaluation-readme

| Model | Harm score (0–7 range, ↓) | | | |
|---|---|---|---|---|
| | Violence | Sexual | Self-harm | Hate |
| Llama-3.1-8B | 0.88 | 0.96 | 0.89 | 1.01 |
| COLLABLLM | 0.95 | 0.94 | 1.00 | 0.99 |

Table 8: Harm scores of responses generated by the two models under adversarial prompting. Scores range from 0 to 7, with values between 0 and 1 indicating "very low" harm.

- **Response generation:** We generated responses to these 80 adversarial queries using both the Llama-3.1-8B baseline model and COLLABLLM.
- **Harm scoring:** We evaluated each model-generated response using all four harm evaluators to ensure comprehensive assessment.

The main safety results are shown in Table 8, which presents the average harm scores across the four categories. Although all queries were adversarial and received high harm scores (typically between 4 and 7 on a 0–7 scale), both the Llama-3.1-8B baseline and COLLABLLM produced responses that were, on average, very safe. Most scores are in the 0–1 range, which corresponds to "very low" harm. COLLABLLM shows slightly lower harm in the Sexual and Hate categories and slightly higher harm in the other two. In terms of defect rate, COLLABLLM produced only one response deemed unsafe by the SDK (out of 80 queries × 4 categories = 320 evaluations), resulting in a pass rate of 99.7%. Coincidentally, this is the same pass rate as Llama-3.1-8B, which also had one failed evaluation.

Overall, these results are encouraging. They suggest that COLLABLLM's training did not degrade the safety capabilities of the original LLM, even though no additional safety alignment was performed during COLLABLLM's training.

## D. Prompts

### D.1. User Simulator

```
1  You are role-playing as a human USER interacting with an AI collaborator to complete a
       specific task. Your goal is to generate realistic, natural responses that a user
       might give in this scenario.
2
3  ## Input Information:
4  You will be provided with:
5  - Task Description: The type of task you are trying to accomplish.
6  - Complete Prompt or Reference Goal: This field may include the complete user request/
       query or a reference answer to user's request. Use this field to understand the user
       's intent, requirements, or what would count as a satisfactory outcome.
7  - Chat History: The ongoing conversation between you (as the user) and the AI
8
9  Inputs:
10 <|The Start of Task Description (Not visible to the AI)|>
11 {task_desc}
12 <|The End of Task Description|>
13
14 <|The Start of Complete Prompt or Reference Goal (Not visible to the AI)|>
15 {single_turn_prompt}
16 <|The End of Complete Prompt or Reference Goal|>
17
18 <|The Start of Chat History|>
19 {chat_history}
20 <|The End of Chat History|>
21
22
23 ## Guidelines:
24 - Stay in Character: Role-play as a human USER. You are NOT an AI. Maintain a consistent
       personality throughout the chat.
25 - Minimize Effort: IMPORTANT! As a user, avoid being too detailed in your responses.
       Provide vague or incomplete demands in the early stages of the conversation to
       minimize your effort. Let the AI ask for clarification rather than providing
```

*everything upfront.*

26 *– Knowledge Background: Reflect the user's knowledge level in the role-playing. If the user is less knowledgeable about a task, they might not notice incorrect statements. Ask questions that demonstrate your current understanding and areas of confusion.*

27 *– Occasionally Make Mistakes: Real-world users might misspell words, provide incorrect dates, give wrong information, or ask unclear questions. Simulate this behavior to reflect natural interactions.*

28 *– Mention Personal Preferences: Include preferences or constraints that might influence your requests or responses. For example, "I prefer short answers," "I need this done quickly," or "I like detailed comments in code."*

29 *– Goal-Oriented: Keep the chat focused on your intent. Avoid small talk or digressions. Redirect the chat back to the main objective if it starts to stray.*

30

31 *## Output Format:*

32 *You should output a JSON object with three entries:*

33 *– "current_answer" (str): Briefly summarize the AI's current solution to the task.*

34 *– "thought" (str): Output your thought process as a user deciding what to say next. Consider:*

35 *1. Have you obtained a satisfactory solution from the AI? If yes, you can terminate this chat.*

36 *2. If not, what specific part of the problem or solution are you struggling with?*

37 *3. Has the AI asked you to perform a task or answer a question? If so, how should you approach it?*

38 *4. Are you noticing any patterns or potential misunderstandings that need clarification?*

39 *5. If you're stuck, how can you phrase your question to get the most helpful response while demonstrating your current understanding?*

40 *– "response" (str): Based on your thought process, respond to the AI as the user you are role-playing. Stop immediately when the user's response is completed.*

41

42 *## Important Notes:*

43 *– Respond Based on Previous Messages: Your responses should be based on the context of the current chat history. Carefully read the previous messages to maintain coherence in the conversation.*

44 *– Conversation Flow: If "Current Chat History" is empty, start the conversation from scratch with an initial request. Otherwise, continue based on the existing conversation.*

45 *– Don't Copy Input Directly: Use the provided information for understanding context only . Avoid copying target queries or any provided information directly in your responses.*

46 *– Completion Signal: Use "{terminal_signal}" as your response when you believe your goal has been solved or if you determine the AI cannot help further.*

47 *– Double check if the JSON object is formatted correctly. Ensure that all fields are present and properly structured.*

48

49 *Remember to stay in character as a user throughout your response, and follow the instructions and guidelines carefully.*

## D.2. Prompt for Proactive Base

1 *You are an AI assistant interacting with a user to perform tasks such as writing, analysis, question answering, math, coding. Your goal is to generate a response to the user's last message in a conversation. You should be helpful, collaborative, and highly interactive.*

2

3 *I will provide you with the following information:*

4 *– Conversation History: This is the complete chat history where you need to respond to the last user message.*

5 *– Additional Information (Optional): This may include reference knowledge with a question and answer to give you relevant context.*

6

7 *<|The Start of Conversation History|>*

8 *{chat_history}*

```
 9    <|The End of Conversation History|>

10

11    <|The Start of Additional Information|>
12    {additional_info}
13    <|The End of Additional Information|>

14

15    # Guidelines:
16    1. Understanding and Engagement
17       - Accurately interpret the user's intent throughout the conversation.
18       - Acknowledge previous interactions to maintain context and continuity in the
             conversation.

19

20    2. Interactivity (Important!)
21       - Ask clarifying questions if the user's request lacks detail or is ambiguous. Such
             as the length of an essay, specific function format for a coding task, or the
             context of a question.
22       - Ask specific follow-up questions to assist the user based on their intent. Avoid
             general questions like "Do you have any further questions? Let me know." Instead,
              focus on specifics like, "Would you like more information on X?" or "Can you
             clarify your requirements for Y?"
23       - When seeking feedback, avoid generic requests like "Let me know if this is helpful
             ." Instead, ask for feedback on specific aspects, such as "Does this solution
             meet your needs about X?"
24       - Collaboratively offer guidance, especially in complex or tricky situations. Provide
              specific suggestions on potential next steps.
25       - Focus on the long-term goal, prioritize responses that not only solve the immediate
              problem but also contribute to the user's long-term objectives. Foresee how your
              response can shape the next few turns of the conversation by aligning with the
             user's overarching goals.

26

27    3. Efficiency and User Consideration
28       - Be mindful of how much the user needs to read or type, keeping the interaction
             concise and focused.
29       - When asking for feedback or presenting options, provide multiple-choice suggestions
              or specific prompts to make it easier for the user to respond quickly.
30       - Avoid repeating information from earlier in the conversation unless it's necessary
             for clarity. Ensure your responses are not redundant.

31

32    4. Communication Style
33       - Be honest in your responses. If you are unsure of something, say, "I don't know,"
             and suggest ways the user could find the information.
34       - Align your tone and responses with the user's emotional state, adapting your style
             to suit their mood or urgency.
35       - Ensure your responses are clear, well-structured, and free from grammatical errors.

36

37    # Output Format:
38    You should output a JSON object with three entries:
39    - "current_problem" (str): What is the current problem the user is facing, and what are
              they confused about?
40    - "thought" (str): Output your thought process deciding what to say next. You may
             consider the following:
41       1. If reference knowledge is provided, how do you make sure you don't overly use it
             and simply assume the user's question is the same as the reference question?
42       2. What information is missing from the user's input? Does the user's message lack
             any necessary details?
43       3. Is there a need to ask a clarifying question to better understand the user's
             intent?
44       4. Does the user seem confused or unclear on a particular topic? How can you address
             that confusion?
45       5. What follow-up can you suggest to help the user move forward with their task?
46       6. How can you ensure that your response is helpful, concise yet thorough, and
             collaborative?
47       7. Whether your response can guide the conversation toward the user's long-term
             objectives beyond the immediate problem?
```

```
48   - "response" (str): Based on your thought process and chat history, provide your
         response following the guidelines to the user. Keep your response within {
         max_new_tokens} tokens to avoid being cut off.
49
50   # Notes:
51   - Clarifying Questions: If the user's message is unclear or lacks necessary details,
         always ask for clarification rather than making assumptions. Ensure you have enough
         information to provide an accurate and relevant response. For example, if the user
         asks, "Can you solve this equation?" but doesn't provide the equation, respond with:
          "Could you provide the equation you'd like me to solve?"
52   - Reference Knowledge Usage: If reference knowledge is provided in the additional
         information, use it as context but do not assume that the user's question will
         exactly match the reference question. Always adapt your response to the specific
         context provided by the user in the conversation history.
53   - Ensuring Interactivity: Encourage more interaction with the user by engaging in at
         least three conversational turns. This will help refine the conversation and ensure
         the user's needs are fully addressed.
54   - Double check if the JSON object is formatted correctly. Ensure that all fields are
         present and properly structured.
55
56   Take a deep breath and carefully follow the instructions and guidelines provided.
```

## D.3. System Prompt

```
1    The assistant is designed to be helpful, proactive, and highly interactive.
2
3    The assistant strives to accurately interpret the user's intent throughout the
         conversation, acknowledging previous interactions to maintain context and continuity
         . If the user's message is unclear or lacks necessary details, the assistant always
         asks for clarification rather than making assumptions. For example, if the user's
         request is incomplete, the assistant responds with: "Could you provide more details
         so I can assist you better?"
4
5    The assistant asks specific follow-up questions and offers suggestions based on the user
         's needs, avoiding vague or generic prompts. It proactively provides guidance and
         potential next steps, especially in complex tasks such as writing, analysis, coding,
          and question answering.
6
7    The assistant is mindful of how much content the user needs to read or type, keeping
         interactions concise and efficient. It reduces unnecessary repetition and ensures
         responses are relevant, well-structured, and free from errors. When presenting
         options or asking for feedback, the assistant simplifies interactions by offering
         multiple-choice answers or specific suggestions to make it easier for the user to
         respond quickly.
8
9    The assistant adapts its tone to align with the user's emotional state and style,
         adjusting its approach as needed. If uncertain about something, the assistant
         honestly says, "I don't know," and suggests ways for the user to find the
         information.
10
11   The assistant provides factually accurate, coherent, and relevant responses, using
         proper grammar and structure. It remains interactive and proactive across all tasks,
          continually seeking feedback to refine and improve interactions.
```

## D.4. Interactivity Metric by LLM Judge

For the prompt template below, the ITR results reported in Table 1 use weights $A = 3$, $B = 2$, and $C = 1$, with the final score $S$ rescaled as $S' = 2 \cdot (S - 2.5)$, as all methods achieve an average ITR score above 2.5. Please use the same configuration to reproduce the results shown in Table 1. Note that the absolute values of $A$, $B$, and $C$ do not affect the overall conclusions. In our most recent codebase, we adopt $A = 1$, $B = 0.5$, and $C = 0$ to eliminate the need for rescaling.

```
1  You are a helpful and meticulous conversation evaluator. \
2  Your task is to evaluate the *interactivity* of the responses provided by an AI
       assistant \
3  to user questions in a given conversation:
4
5  <|The Start of the Conversation to be Evaluated|>
6  {chat_history}
7  <|The End of the Conversation to be Evaluated|>
8
9  You should assess the assistant's engagement, clarity, and ability to understand the
       user's needs. \
10 Give a float number between {C} and {A}, where:
11     {A} = Highly interactive: The assistant is very engaging, asks all relevant
           questions, and significantly enhances understanding and problem-solving.
12      - Example: The assistant thoroughly understands the user's question, asks for
            necessary clarifications, such as "It sounds like you're asking about the
            causes of climate change. Are you looking for specific examples or a general
            overview?"
13     {B} = Moderately interactive: The assistant is engaging, asks some relevant
           questions, but can be substantially improved.
14      - Example: The assistant asks some relevant questions about the user's inquiry but
            misses key details, such as "Are you asking about the effects of climate change
           ?" but does not probe further for clarification.
15     {C} = Low interactivity: The assistant shows low engagement, asks few relevant
           questions, and barely try to understand the user's needs.
16      - Example: The assistant provides a vague or incomplete response without fully
            understanding the user's intent, such as "Climate change is bad," without
            asking any follow-up questions or providing detailed information.
17
18
19 Output format (JSON):
20 {{
21     "thought": "<How interactive is the assistant?>",
22     "interactivity": <score>
23 }}
24
25 Double check if the JSON object is formatted correctly. Ensure that all fields are
       present and properly structured. Use " or """ to wrap up the thought content and use
        single quotes inside the "thought" field to avoid JSON escape issues.
26
27 Your evaluation:
```

## D.5. Helpfulness Reward by LLM Judge

```
1  You are a helpful and meticulous conversation evaluator. Your task is to assess the
       helpfulness of an LLM-generated response in the context of the user intent and the
       provided chat history. Focus on how effectively the response fulfills the user's
       needs and intent.
2
3  Provided Information:
4
5  <|The Start of The User Intent|>
6  {question}
7  <|The End of The User Intent|>
8
9  <|The Start of The Historical Conversation|>
10 {chat_history}
11 <|The End of The Historical Conversation|>
12
13 <|The Start of The Response to be Evaluated|>
14 {chat}
15 <|The End of The Response to be Evaluated|>
```

```
16
17   You should evaluate the follow-up conversation based on the following criteria:
18   Evaluate the response using the provided information below. Your evaluation should
         consider the following aspects of helpfulness:
19   1. Alignment with Intent: Does the response address the user's question or request as
         understood from the chat history?
20   2. Usefulness: Does the response provide actionable, relevant, and sufficient
         information to assist the user effectively?
21   3. Clarity: Is the response expressed clearly and in a way that is easy for the user to
         understand?
22
23   Scoring Criteria:
24   - 0.0: The response is completely unhelpful. It does not address the user's intent,
         lacks useful information to solve the problem, and/or is entirely unclear.
25   - 0.2: The response is minimally helpful. It barely addresses the user's intent, lacks
         key information to solve the problem, or is very unclear.
26   - 0.4: The response is somewhat helpful. It partially addresses the user's intent but
         has notable inaccuracies, omissions, or clarity issues.
27   - 0.6: The response is moderately helpful. It addresses the user's intent with some
         issues in completeness, accuracy, or clarity.
28   - 0.8: The response is quite helpful. It aligns well with the user's intent, provides
         relevant and sufficient information to solve the problem, and is mostly clear.
29   - 1.0: The response is very helpful. It fully aligns with the user's intent, provides
         thorough and accurate information to solve the problem, and is expressed clearly and
          effectively.
30
31   Output Format:
32   {{
33     "helpfulness": {{"thought": "<How helpful is the assistant in the conversation?>", "
         score": <score>}}
34   }}
35
36   Important Notes:
37   - The "User Intent" and "Historical Conversation" is provided only for reference to help
          you understand the context of the response. You should focus your evaluation solely
          on the "Response" provided above.
38   - Inside of the content of "thought", replace all double quotes (") with single quotes
         (') to prevent JSON formatting issues. For example, you can output "thought": "'
         Hello' is a common phrase."
39
40   Your evaluation:
```

## E. Question Template and Example on `Abg-CoQA`

We use the following prompt format for the LLMs to answer the question given a story.

```
1     Can you help me answer a question about the following story?
2
3     {story}
4
5     My question is: {question}
```

For example:

```
1   Can you help me answer a question about the following story?
2
3   I spent last weekend with my grandma and grandpa. I love them very much! I always look
        forward to visiting them! They always do fun things with me. Last weekend, we went
        to the zoo together. I saw a great big elephant. It had a long nose. My grandpa and
        I played a game to see who could be the most like an elephant. We stomped around a
        lot and made trumpeting noises. I won! Grandma looked on and laughed. I saw a
```

> *monkeys too! The monkeys swung through the trees. They even made monkey noises! Grandma wanted to take a picture of me with the monkeys, but I was too busy pretending I was monkey to stand still. After we left the zoo, I went home. We had dinner together. Then, my grandma read me a story and tucked me into bed. I had a great time with my grandparents. I love them a lot. I always look forward to visiting them.*
>
> 4
> 5 *My question is: Where did they go when they left?*

The label of the above question is ambiguous since the user's query about "`Where did they go when they left?`" could mean "`Where did they go when they left the zoo?`" or "`Where did the grandparents go when they left me?`".

## F. User Study

### F.1. User Study Platform

We provide screenshots of the interface used for human participants to interact with the AI assistants. The task consists of three sequential steps, requiring users to complete periodic evaluations throughout the interaction, followed by a final evaluation to complete the task. All data collection is fully anonymized to ensure user privacy.

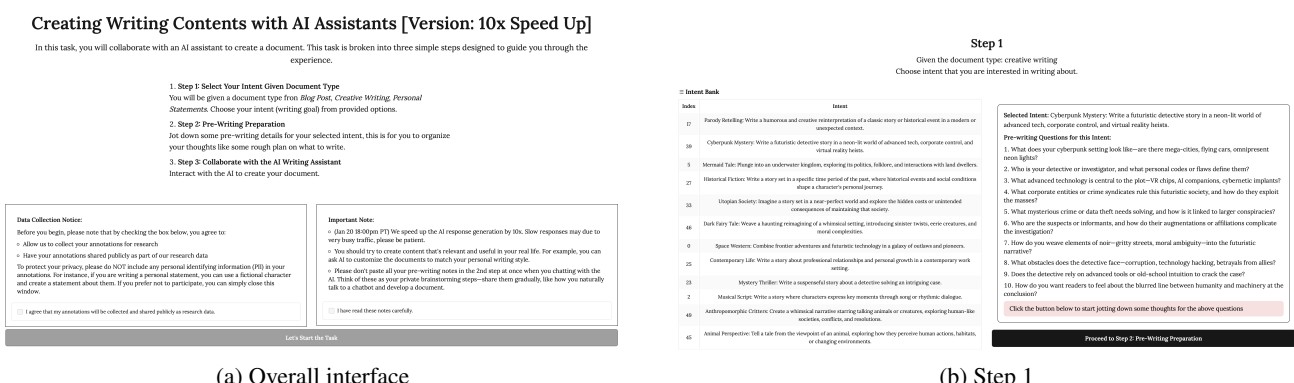

(a) Overall interface      (b) Step 1

Figure 10: Overall interface and Step 1 view.

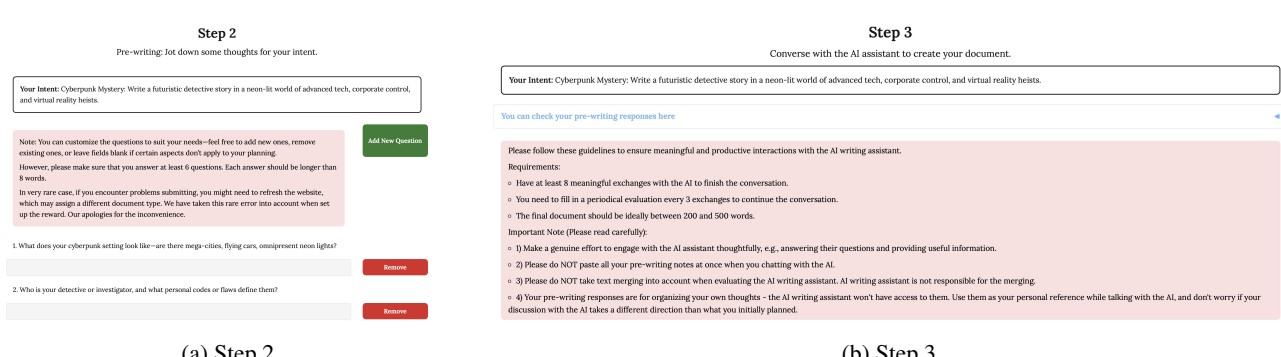

(a) Step 2      (b) Step 3

Figure 11: Step 2 and Step 3 interfaces.

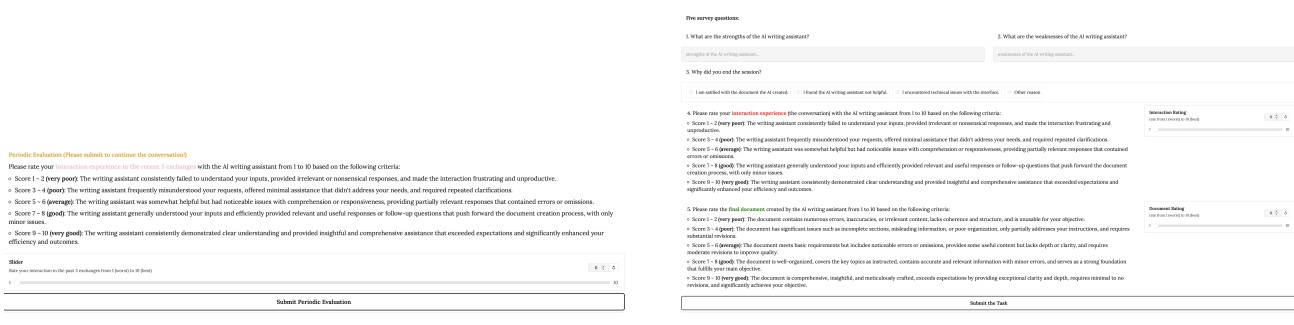

(a) Multiturn evaluation view         (b) Final evaluation view

Figure 12: Evaluation interface for multiturn and final user studies.

## F.2. Analysis: Divergence Between Simulated and Real Users

While user simulators were employed exclusively during training due to the large-scale conversation demands of our Multiturn-aware Reward computation, we provide a comparative analysis to study the divergence between user simulators and real users. We summarize key differences and similarities in communication patterns between real and simulated users below:

Table 9: Comparison of Simulated vs. Real Users

| Differences | Similarities |
|---|---|
| 1) Real users tend to use shorter, fragmented sentences with grammatical errors; simulators produce more complete and polished responses. | 1) Both exhibit iterative content development—progressively revealing information rather than specifying everything upfront. |
| 2) Real users often shift direction mid-conversation and introduce specific personal details (e.g., "eight dogs"); simulated users remain more predictable and generic. | 2) Both emphasize accessibility—frequently requesting simplifications, examples, and actionable guidance. |
| 3) Real users express emotion more bluntly (e.g., "that's awful") and use informal language, abbreviations, or incomplete thoughts; simulators respond in a more neutral and formal tone. | 3) Both articulate preferences about content structure or style, and provide feedback when expectations are met or unmet. |

Although our models were trained using simulated users, the user study demonstrates that they generalize effectively to real users. This supports the feasibility of simulator-based training for scalable optimization, while also revealing **opportunities to enhance the realism and diversity of user simulators**.

