# OpenReview forum: "CollabLLM: From Passive Responders to Active Collaborators"
_ICML.cc/2025/Conference — ICML 2025 oral_

### Official Review · Reviewer_mUmH · 2025-02-27

**Overall Recommendation:** 4

**Summary:**

While state-of-the-art Large Language Models (LLMs) trained with RLHF are good at following the instructions from users, this paper argues that they are often ``passive responders'' where they only passively respond to ambiguous or open-ended user requests. To address this limitation, this paper proposes to train LLMs with multi-turn aware utility through a conversation-level reward and a forward sampling process. The conversation-level reward is composed of an extrinsic reward of task completion and intrinsic reward that prioritizes user experiences. Experiments have shown that in three simulated tasks, CollabLLM (trained with either PPO or DPO) is able to achieve better performances compared to prompting baselines. A large-scale user study is also carried out and it is shown that CollabLLM can indeed enhance the user satisfaction over multiple turns.

**Claims And Evidence:**

In general, claims in the paper are well supported through empirical evidence, both through simulated experiments of three tasks and a conducted user study with Mechanical Turks. I particularly appreciate the inclusion of the user study with more than 200 participants that show the positive generalization from model-simulated user to real users.

**Essential References Not Discussed:**

Would be nice to have a comparison with other works that make use of user-simulators to improve LLMs, such as Zero-Shot Goal-Directed Dialogue via RL on Imagined Conversations (https://arxiv.org/abs/2311.05584).

**Experimental Designs Or Analyses:**

The experiment designs make sense in general, and I appreciate the ablation of the effect of reward mechanisms in Figure 4 and Figure 6, and the zero-shot generalization experiment in Table 2. However, it seems strange why a different model GPT4-o is used as the user simulator compared to the original model Llama-3.1-8B-Instruct. It would nice to have some additional discussions in terms of the effect of using a stronger model as the user simulator, and I would be curious to see if it is possible for the model to improve with "self-play" without relying on a stronger model as a user simulator.

**Methods And Evaluation Criteria:**

Do proposed methods and/or evaluation criteria (e.g., benchmark datasets) make sense for the problem or application at hand?

Yes, the proposed evaluations are well-suited for the problem.

**Other Comments Or Suggestions:**

See above

**Other Strengths And Weaknesses:**

See above

**Questions For Authors:**

My main questions are mentioned in the previous responses and I summarize them below:

1) While I understand that real-user conversations can be hard to obtain, it may strengthen the paper and better situate the paper in the related literature if the authors can provide quantitaive comparisons with prior "Multiturn training for LLMs" as mentioned in the related works section.

2) It would be nice to add discussions explaining the difference of this paper from other prior literature that uses LLMs as user simulators, e.g. ttps://arxiv.org/abs/2311.05584.

3) It would be helpful if the authors can provide some insight in terms of the limitations of applying LLMs as user simulators. In my experience, I found instruction-tuned LLM as user simulators tend to be overly agreeable and always consent to the requests while real users can be harder to deal with. Curious if such issues are also observed in the experiments.

4) Would be nice to se

**Relation To Broader Scientific Literature:**

I find a detailed discussion of the paper's contribution with the broader literature missing from the main text. Although Appendix B mentions related works, I believe that it is important to address the connection of the paper to the prior literature in the main text. Additionally, in the related works section, the paper mentions that "However, these methods primarily rely on post-hoc trajectory-level data, learning from observed conversations rather than explicitly modeling the causal effect of individual responses on long-term task success (see Appendix A for further details).", but the relative advantage of learning from real-user conversations and explicitly modeling the causal effect of individual responses seem to be missing from the paper as far as I can tell. While I understand that real-user conversations might be hard to obtain, it would be valuable if the authors can provide some quantitative comparisons with the methods used in the prior literature.

**Theoretical Claims:**

No theoretical claims made in the paper.

---

> ### Author Rebuttal · Authors · 2025-04-01
>
> We thank the reviewer for their approval and rigorous comments! Here we address the remaining concerns:
>
> ---
>
> ### **[Experimental Designs] "It would be nice to have additional discussions about using a stronger model as the user simulator.”**
>
> Great catch! A user simulator should follow the language style of previous user turns while exhibiting typical user behaviors like evolving needs and limited background knowledge. This requires LLMs to role-play users with **basic understanding of real-world user traits and follow instructions effectively** [1,2,3].
>
> Though not motivated by self-training, we initially tried using `Llama-3.1-8B` as our user simulator to reduce latency. Unfortunately, it performed poorly, frequently getting "confused" and solving problems as an assistant instead of a user. This observation is the same even for `Llama-3.1-70B`.
>
> We think this raises an interesting research problem - *while we have increasingly superior LLM assistants trained to solve problems, we lack user models that learn from real-world user behaviors*. Building better user models can be valuable running simulations for real-world applications.
>
> Therefore, "self-play" was difficult to implement within the scope of this work, but this points to potential future work. We've added discussion in future work directions.
>
> ---
>
>
> Moreover, to reviewer’s Question \#3 - **“It would be helpful if the authors can provide some insight in terms of the limitations of applying LLMs as user simulators.”**. We discussed this in our response to `Reviewer 2AZE` (please see the section's beginning), given space constraints. We've added the discussion in the paper to provides insights.
>
> ---
>
>
> ### **[Relation To Broader Scientific Literature]**
>
> **(1) “(add) prior literature in the main text.”**
>
>
> Yes we agree. We will and should have space in the final version to accommodate the Related Work section in the main text!
>
> ---
>
>
> **(2) “it would be valuable if the authors can provide some quantitative comparisons with prior "Multiturn training for LLMs”**
>
> Thanks for this suggestion! Previously we looked into MTPO (Shani et al. 2024), but unfortunately we didn’t find an existing implementation. Another relevant method in the category, ArCHer (Zhou et al. 2024), is a hierarchical multiturn RL approach that requires the training of three LLMs: 1) utterance-level Q-function and 2) utterance-level value function, and 3) a token-level actor that maximizes the prediction of the Q-model, which exhibits high complexity in our setup. Moreover, this contributes to a more task-specific training with learning the token-by-token policy within each turn, while CollabLLM offers a more intuitive and generalizable method for multiturn interaction with a single model.
>
> ---
>
>
> ### **[Essential References Not Discussed] "(compare with) works that use user simulators to improve LLMs”**
>
> We further added the following content in the related work section:
> ```
> Recent works employ user simulators to enhance dialogue systems [4,5] and LLMs [6,7,8]. Tseng et al. improve both dialogue systems and simulators through reinforcement learning during their interactions. Recently, Hong et al. leverage LLMs to create diverse synthetic dialogues with varying user personas, then train smaller dialogue models to optimize conversational outcomes. CollabLLM differs in leveraging user simulators in forward sampling to account for long-term effect in both offline and online training.
> ```
>
> It is also worth mentioning that our main contribution comes from the key intuition in making the model be aware of future outcomes and prioritize responses with higher long-term impact. The other components are fundamentally aimed at computing long-term effects, with the use of a user simulator for forward sampling being just one, albeit minor, contribution of ours.
>
> ---
>
> ## **Summary**
>
>
> We sincerely hope our answers mitigates your concerns about (1) the user simulator models and their difference between real users (2) the comparison with works within "Multiturn training for LLMs" and works using user simulators and.
>
> Given our answers, we would very much appreciate it if you could reconsider your evaluation. Thank you very much!
>
>
> ---
>
>
> ## **Reference**
>
> [1] Park et al. Generative Agent Simulations of 1,000 People. ArXiv.
>
> [2] Wang et al. User Behavior Simulation with Large Language Model-based Agents. ArXiv.
>
> [3] Yoon et al. Evaluating Large Language Models as Generative User Simulators. NAACL 2024.
>
> [4] Shi et al. How to Build User Simulators to Train RL-based Dialog Systems. EMNLP 2019.
>
> [5] Tseng et al. Transferable Dialogue Systems and User Simulators. ACL-IJCNLP 2021.
>
> [6] Hong et al. Zero-Shot Goal-Directed Dialogue via RL on Imagined Conversations. ArXiv.
>
> [7] Hu et al. Unlocking the Potential of User Feedback: Leveraging Large Language Model as User Simulator to Enhance Dialogue System. CIKM 2023.
>
> [8] Faltings et al. Interactive Text Generation. EMNLP 2023.

---

### Official Review · Reviewer_YkB9 · 2025-03-03

**Overall Recommendation:** 4

**Summary:**

COLLABLLM is a new training framework designed to improve multi-turn human–LLM collaboration. Its core idea is to simulate a collaborative conversation setup where a Multiturn-aware Reward (MR) function estimates the long-term impact of model’s responses, rather than focusing solely on immediate single-turn outcome (as in standard RLHF).

Main Contributions:
-Multiturn-aware Rewards (MR): A conversation-level reward function that encourages the LLM to seek and incorporate additional context or clarification from users if it improves overall task success.
-New Multi-turn Interaction Benchmark: which covers 3 challenging tasks related to document editing, coding, and mathematics.
-COLLABLLM outperforms base (or prompt-engineered) baselines on 3 test sets by boosting task accuracy by 18.5% and interactivity by 46.3%, as judged by LLM evaluators. In a large-scale user study with 201 Amazon Mechanical Turkers, COLLABLLM also increases user satisfaction by 17.6% and saves 10.4% of user time compared to baselines.

**Claims And Evidence:**

Yes they are (although i have concerns/reasons to reject expressed in the specific strength/weeknesses section)

**Essential References Not Discussed:**

They are discussed but in the supp material

**Experimental Designs Or Analyses:**

Yes i checked:
-simulated experiments
-crowdsourcing study

**Methods And Evaluation Criteria:**

Yes they are  (although i have concerns/reasons to reject expressed in the specific strength/weeknesses section)

**Other Comments Or Suggestions:**

Typo:
-caption of fig 2: Figure 2: Real examples from COLLABLLM and non-collaborative LLM fine-tuing
=> fine-tuNing

**Other Strengths And Weaknesses:**

reason to accept
-A well-designed framework aimed at improving multi-turn human–LLM collaboration, supported by  experimental evaluation.
-Introduction of a new benchmark specifically for evaluating multi-turn interactions.
-An in-depth case study that goes beyond accuracy metrics, providing deeper insights into model behavior and interaction quality.
-Proposed approach implicitly addresses (and casually solves?) the question clarification problem by optimizing for long-term goal achievement, naturally encouraging the model to seek more clarifications when needed, as demonstrated in the paper.

reasons to reject
-The improvements on simulated experiments of tab 1 (35% to 36-38% BLEU, 12.5 to 15) are small between prompt engineering and the proposed method (using PPO or DPO), raising doubts about real impact. With overall small performance improvement and a model size ≤8B parameters, the validation of the method is not 100% convincing to me. What would be the topline obtained w/ gpt4-o and the same prompt engineering for instance ?
-It's unclear whether the improvements stem specifically from the multi-turn-aware reward (with w>0, regardless of whether the reward is based on helpfulness, intrinsic, extrinsic, or a mix) or from the reward modification itself (replacing helpfulness with extrinsic + intrinsic rewards). or alternativelly is it the interaction between both factors that drives the gains?

**Questions For Authors:**

-MediumDocEdit-Chat: task performance is evaluated using BLEU, which measures similarity between the extracted document and the original articles.
How is the document extracted? It's unclear what exactly is being generated here.
Is BLEU the right metric for this task? Why not also use LLM judges for a more qualitative assessment?
-Interactivity (ITR): engagement is evaluated using an LLM judge (Claude-3.5-Sonnet) with scores rescaled to [0,1]
But how exactly is this scoring performed? The methodology needs more clarity.
-Figure 4: why does ITR performance decrease when the forward sampling window size increases from w=2 to  w=3? This behaviour seems counterintuitive to me. What could explain it?
-What about optimizing helpfulness (as assessed by the LLM evaluator) using w>0? Is it feasible? If so, why was this approach not explored?

**Relation To Broader Scientific Literature:**

Paper is well positioned related to the scientific litterature, although i'm not a specialist of multi-turn human–LLM collaboration.

**Theoretical Claims:**

There are no strong theoretical claims in this paper

---

> ### Author Rebuttal · Authors · 2025-03-31
>
> We appreciate the reviewers' extensive and thoughtful outputs! We'll address each comment below:
>
> ---
>
> ### **[Other Weaknesses] "The improvements (over prompt engineering) on simulated experiments of Tab 1 are small… What would be the topline obtained w/ gpt4-o + prompt engineering?”**
>
> Thanks for the comment! We'd like to provide the following justification:
>
> - **Task-specific Performance upper bound of improving collaborative strategies:** The training of CollabLLMs does not necessarily provide more task-specific knowledge, rather, the primary goal is to explore the best collaboration strategies for the models to understand user requests and deliver its internal knowledge. Therefore, we believe an upper bound constraint exists for improvement we gain from improving collaboration strategies.s
>
> - **Challenging Tasks:** In particular on MediumDocEdit-Chat, the task itself is challenging with a large generation space for writing the blogs, making the BLEU metric hard to improve.
>
> Given the first reason, comparing with gpt-4o (in general, comparing between models with different base models) may not be fair, as it's difficult to isolate effects from stronger knowledge versus better collaborative strategies. There might also be data contamination risks with gpt-4o, which likely trained on internet-scale data possibly including our datasets. However, we're willing to run testing if you're curious about the results.
>
> ---
>
> ### **[Other Weaknesses + Questions] "Where do the improvements stem from? Multiturn-aware reward design, or reward modification, or interaction between both factors?”**
>
> This relates to one of the reviewer’s questions - `"What about optimizing helpfulness using w>0?"`. For validation, we extended helpfulness, intrinsic, and extrinsic metrics to $w=1,2,3$ in model training, following the same setting as Figure 4.
>
> Please see visualized results at: https://anonymous.4open.science/r/collab-llm/images/ablation.png. Previously, we see helpfulness as a type of intrinsic reward. Here, we found applying helpfulness alone doesn't work well on all metrics, especially as it encourages lengthy responses. For the other metrics, increasing $w$ generally benefits the performances corresponding to the metric. For example, applying extrinsic rewards improves the BLEU score, while the ITR and token amount underperforms Extrinsic+Intrinsic adopted in CollabLLM.
>
> Notably, the design of extrinsic and intrinsic rewards is independent of the key design in MR function, which highlights the estimation of a response’s long-term effect via forward samplings. In fact, one can apply multiple intrinsic rewards including helpfulness and extrinsic rewards in the MR.
>
> We hope the additional ablation clarifies the source of improvements, we have changed `Figure 4` with this more comprehensive study.
>
> ---
>
> ### **[Questions] Clarify the experiments on MediumDocEdit-Chat**
>
> **(1) “How is the document extracted?”`**
>
> The document is extracted by prompting an LLM to extract the final written content after multiple conversational turns between the user simulator and the model, then comparing this generated document with the original Medium article using BLEU score to evaluate similarity. We added this description to the experiment setup.
>
> **(2) “How exactly is this scoring for ITR metric performed?”`**
>
> Please see `Appendix D.4`, starting from `Line 915`, we provide full prompts to produce ITR.
>
> **(3): “Is BLEU the right metric for this task? Why not also use LLM judges?”`**
>
> Yes we can use LLM judges, but since we already have human evaluation on document quality in the user study, which, we believe, is perhaps more convincing to assess the document quality.  We're happy to reevaluate if the reviewer thinks otherwise.
>
> ---
>
> ### **[Questions] "Figure 4: why does ITR performance decrease from $w=2$ to $w=3$?”**
>
> Great catch! Following the MR formulation in `Eq. 1`, with increasing $w$, each model response's effect estimation on the final goal should be more accurate, and ITR performance should improve. However, for scalability, we conduct Monte Carlo sampling for future conversations with sample size fixed at 3 `(Appendix C.2)`, inevitably introducing estimation errors. This explains potential fluctuation with increasing $w$. We added this interpretation to the ablation section. Thanks for this question!
>
> Lastly, we have fixed the typo identified by the reviewer.
>
> ---
>
> ## **Summary**
>
> We deeply appreciate your detailed and thoughtful comments. We hope our answers address concerns about (1) significance of improvements, (2) source of improvements, and (3) clarity about evaluation and ablation results. Please let us know if you have more questions! Thank you again!

---

> > ### Comment · Reviewer_YkB9 · 2025-04-03
> >
> > Thank authors for having addressed my questions and concerns. I have nothing to add here and overall this confirms my positive feedback on the paper. I still think having an upper bound gpt4-o topline (comparing with gpt-4o) would be informative though...

---

> > > ### Author Response · Authors · 2025-04-07
> > >
> > > Thanks to Reviewer YkB9 for the comment and acknowledgement. For sure, we are happy to provide the reference results running gpt-4o with proactive prompting: https://anonymous.4open.science/r/collab-llm/images/gpt-4o-reference.png
> > >
> > > - For **task-specific metrics**, gpt-4o achieves the best results on math and coding tasks, which is expected since gpt-4o has exhibited much stronger knowledge from pretraining compared to `Llama-3.1-8b`. However, the performance of gpt-4o on the document editing task is particularly low, which emphasizes the positive impact of our multiturn-aware training on open-ended tasks even when compared to a much stronger model.
> > >
> > > - For **number of tokens**, gpt-4o generates 28.9% more content when compared with `Proactive Base`, and 51.3% more content when compared with our `Online DPO models`. We observe that the actual generation from gpt-4o, e.g., on the document editing task, is extremely extensive, especially when the user simulator didn't specify the length.
> > >
> > > - For **interactivity**, gpt-4o are slightly better than CollabLLMs on the math and coding tasks, while the interactivity is lower than CollabLLMs on document editing tasks.
> > >
> > > We hope this reference provides more information (also we apologize for replying a bit late). We have added the results to the paper.

---

### Official Review · Reviewer_2AZE · 2025-03-18

**Overall Recommendation:** 4

**Summary:**

This paper studies how to enhance human-AI collaboration by improving multi-turn conversations. Concretely, authors propose a learning framework CollabLLM that uses a reward function aware of multi-turn setup in reinforcement finetuning. This multiturn-aware reward takes account of both task performance and user satisfaction, and is proved empirically effective in a few simulated environments including text editing, code generation and math reasoning.

**Claims And Evidence:**

- This work addresses a key limitation of existing LLMs: the tendency to generate single-turn responses without actively engaging in clarifying or guiding user intents.
- The proposed multiturn-aware reward function is an interesting contribution, as it incorporates both extrinsic task success metrics and intrinsic user experience factors (e.g., excessive tokens to read and write).
- This work thoroughly evaluates CollabLLM across multiple tasks, showing substantial improvements in task success and user engagement with simulated users. Additional human evaluation with 201 crowd worker participants provides empirical validation beyond automated benchmarks, showing increased user satisfaction and reduced time spent on tasks.
- The ablation section provides useful insights into the importance of forward-looking strategies in reinforcement learning.
- This paper is very well written.

**Essential References Not Discussed:**

N/A

**Experimental Designs Or Analyses:**

Experimental design and analysis sections look sound and thorough.

**Methods And Evaluation Criteria:**

Three multiturn interaction benchmarks are proposed, covering document editing, code generation, and math problem-solving. The evaluation criteria is diverse, including measurements on task accuracy, interactivity, user satisfaction, user efforts etc.

**Other Comments Or Suggestions:**

N/A

**Other Strengths And Weaknesses:**

- A comparison of the potential divergence of simulated user and human user during training would further strengthen this work, as prompt-defined simulated LLM users could be substantially biased.
- Multiturn-aware reward function is intrinsically hard to define for ambiguous tasks, which limits its applicability.

**Questions For Authors:**

Any idea on the computational expensive aspect of the forward sampling strategy? This seems nontrivial especially for long conversations.

**Relation To Broader Scientific Literature:**

The discussion around the suboptimal performance in handling multi-turn interactions is well-motivated and supported by literature. The insight of using multiturn-aware reward and forward sampling strategies is shown to be effective, and seems generalizable to other tasks.

**Theoretical Claims:**

N/A

---

> ### Author Rebuttal · Authors · 2025-03-31
>
> We sincerely appreciate your approval and insightful comments! We further address your comments:
>
> ---
>
>
> ## **Comment 1: "A comparison of the potential divergence of simulated user and human user during training would further strengthen this work"**
>
> Good point! We agree that prompt-defined user simulators could be biased. Due to the large-scale forward conversations needed for computing Multiturn-aware Rewards, we only used the user simulator during training. While not meant for training, in the user study, we have collected conversations between real users and our models.
> Here we provide some insights on feature differences and similarities between prompt-defined user simulators and real users' responses.
>
> **Differences:**
> - *Real users communicate with shorter, more fragmented sentences often containing grammatical errors, while simulated users typically use more complete sentences.*
> - *Real users frequently change direction mid-conversation and add highly specific personal details (like "eight dogs"), while simulated users are more predictable.*
> - *Real users express emotional reactions more bluntly ("that's awful," "sounds pretentious") and use more casual language patterns with abbreviations and incomplete thoughts compared to simulated users.*
>
> **Similarities**:
>
> - *Both exhibit iterative content development patterns - gradually revealing requirements rather than providing complete information upfront.*
> - *Both prioritize accessibility - consistently requesting simplification of complex topics, actionable advice, and concrete examples that make information more understandable.*
> - *Both express preferences about content structure and style, and acknowledge when content meets or doesn't meet their expectations.*
>
> We train models by interacting with simulated LLM users, while conducting user studies to evaluate model performance with real users as a test of generalization. The generalization is validated from the experimental improvements. With more resources, it would be interesting to further explore the sensitivity of the impact of users on the trained models.
>
> We are glad that the reviewer raises this interesting question. We have added these insights to `Appendix: F`, as well as adding the key gap in `Section 6: Real-world User Study`. Hopefully these could shed light into future human-centered LLM training.
>
>
> ---
>
>
> ## **Comment 2: "Multiturn-aware reward function is intrinsically hard to define for ambiguous tasks"**
>
> Thanks for the comment! Optimizing for ambiguous tasks could be hard even in single-turn settings, in addition to the multiturn settings that we study.
>
> In general, on ambiguous tasks, such as recommendation or consultant, one mitigation is to use LLM Judges whose inputs contain the task definition. The assumption is that, since LLMs are powerful at reasoning, they are fairly good at telling if the task completion is good or not. Empirically, this has been commonly adopted in evaluation and benchmarking.
>
> For our Multiturn-aware Reward function, we incorporate both extrinsic and intrinsic rewards, where the intrinsic rewards (interactivity and efficiency) should be applicable to most applications. For extrinsic/task-specific metrics, the same design discussed above can be applied over the future conversations.
>
> We have added this discussion to our main paper!
>
> ---
>
>
> ## **Question 1: "the computational expensive aspect of the forward sampling strategy?"**
>
> Thanks for this question! In online training, the computational overhead in forward sampling comes from 1) generation from the policy model and 2) generation from the user simulator.
>
> - For (1), the computational overhead and cost are fairly low since we have integrated vLLM [1] in model inference.
> - For (2), we use gpt-4o-mini as the user simulator where we expect user responses to be concise, i.e., the number of output tokens to be small.
>
> We compute the average statistics over 100 future conversations when w=1,2,3 on MediumDocEdit-Chat, the document editing task, which has the maximum computational overhead among the three tasks. Please see the table in https://anonymous.4open.science/r/collab-llm/images/cost.png
>
> We have added this information to `Appendix C.3: Computational Cost During Training`, and hopefully this provides clear details.
>
> ---
>
>
> ## **Summary**
>
> We thank the reviewer for the interesting questions and comments. We hope our responses can alleviate the concerns on 1) user simulators, and 2) applicability in ambiguous tasks. We also provide more details about computational expenses, as well as improving our manuscript accordingly. Please don't hesitate to let us know if you have more questions!

---

### Official Review · Reviewer_Qxpu · 2025-03-18

**Overall Recommendation:** 4

**Summary:**

Existing fine-tuning techniques for LLMs, such as Reinforcement Learning from Human Feedback (RLHF), primarily maximize the reward for immediate and single-turn responses. However, real-world users often reveal their intents or preferences until later interactions; thus, to streamline their interaction with users and improve user satisfaction, LLMs must be able to actively guide users to clarify and refine their intents throughout the multi-turn conversation. This paper proposes ColabLLM, a novel training framework that encourages LLMs to collaborate with humans in multi-turn conversations. The collaborative simulation module of ColabLLM samples future conversations with users to estimate how the LLM response would impact future turns.  This long-term impact, termed Multiturn-aware Reward (MR), evaluates responses based on both task-specific success and efficiency to assess the multi-turn collaboration quality. Once this MR is computed, ColabLLM employs established RL algorithms to fine-tune the backbone LLM. In addition, the paper releases three multiturn datasets across diverse domains - collaborative document editing, coding problem assistance, and multiturn problem solving - to fine-tune and evaluate LLMs' multiturn conversational capabilities.

**Claims And Evidence:**

C1. ColabLLM encourages LLMs to collaborate with human users in multiturn conversations -> lacks evidence or clarity
- It is clear that multiturn data are collected with an LLM that is prompted as a user simulator. On the contrary, it is rather unclear how the multiturn reward obtained with multiturn data effectively encourages collaboration.

C2. The reward design of ColabLLM aligns with causal effect estimation. -> somewhat convincing
- Could the authors elaborate more on their claim in lines 685-687 of the Appendix "existing methods primarily rely on post-hoc trajectory-level data, learning from observed conversations rather than explicitly modeling the causal effect of individual responses on long-term task success?" Is this because the existing methods do not have a user simulator, and thus they lack the ability to probe the long-term impact of LLMs responses? Then, how do existing methods create post-hoc trajectory-level data? How are these different from the data created by ColabLLM? These distinctions would make the related works section more comprehensive and improve the paper's quality as a standalone academic paper.

C3. Three datasets for fine-tuning and evaluating LLMs on multiturn conversations are proposed -> lacks evidence
- Please correct me if I am wrong, but I could not find supplementary materials or anonymized links to these datasets. While the authors provide samples in the Appendix, I would have preferred to see the full datasets to get a clearer picture of the dataset quality and quantity.

**Essential References Not Discussed:**

N/A

**Experimental Designs Or Analyses:**

The experimental designs and analyses are valid.

**Methods And Evaluation Criteria:**

The proposed method is technically sound. The evaluation criteria are aligned with the convention in this field.

**Other Comments Or Suggestions:**

As collecting real data is expensive, and it is commonly believed that most of the data available at hand (ex) via internet crawling) have been exhausted during LLM pre-training, utilizing LLM-generated data to improve performance on downstream tasks or to encourage a certain behavior is gaining popularity. It appears that this work shares this philosophy of self-training, and has made clever modifications to tailor it specifically for multi-turn conversation capabilities. Therefore, the methodology, while described using fancy technical terms, such as "forward-looking strategies" or "user simulator," may not necessarily be as novel as the authors claim them to be. For instance, when explained in plain terms, estimating forward-looking strategies with a user simulator is more or less the same as generating more realistic multi-turn data with an LLM that is prompted to behave like a human user. Therefore, the motivation and design principle behind the methodology must be better conveyed to showcase that this work goes beyond simply engineering and re-designing the self-training framework for the purpose of multiturn conversations and does indeed reveal an unknown or discussion-worthy application of LLM-backed data generation. If this concern, raised in the Claims and Weaknesses section, could be addressed, I am willing to raise my score to recommend acceptance of the paper.

**Other Strengths And Weaknesses:**

Strengths
- The problem is well-motivated. Improving the LLMs' multiturn conversational capability is an important problem.
- The proposed method, which relies on data generation with a user simulator and multi-turn reward design, is technically sound.
- The paper additionally introduces three public benchmarks for multi-turn conversation research.
- The results are strong, and the authors compare the proposed method against strong baselines.
Weaknesses (already mentioned in the Claims and Evidence section)
- How the proposed method encourages collaborative behavior could be better discussed.
- The cause-effect estimation with the user simulator could use some clarification.

**Questions For Authors:**

Please refer to the above section.

**Relation To Broader Scientific Literature:**

This was included in the summary.

**Theoretical Claims:**

The claim on the cause-effect estimation wiht ColabLLM is unclear; this concern was raised above.

---

> ### Author Rebuttal · Authors · 2025-03-31
>
> We appreciate your approval and useful feedback! Here we address your concerns:
>
> ---
>
> ## **[Claims And Evidence] "How does the multiturn reward effectively encourage collaboration?"**
>
> - At the **methodology** level, Multiturn-aware Reward (MR) encourages collaboration by accounting the long-term effect of each response in future interactions. A more collaborative LLM should achieve higher extrinsic and intrinsic rewards within a fixed number of turns, corresponding to task completion and efficiency/engagement. `=> clarity`
>
> - At the **data** level, since the data is generated from applying MR, supervised training on the multiturn data encourages the model to replicate the behavior. MR is also used for online training, which reinforces the model to optimize MR based on current generations. `=> clarity`
>
> - At the **experiment** level, we show that in simulated environments and real-world user study, CollabLLMs achieve best collaboration with humans in the collaborative aspects we considered. `=> evidence`
>
> ---
>
> ## **[Claims And Evidence] Could the authors elaborate more on lines 685-687.**
>
> We apologize for this concise statement and provide the following clearer explanation, and we will update the paper accordingly:
>
> - In plain language, our MRs provide **turn-level** signals as the long-term effect of each model response, while existing methods such as MTPO [1] applies **trajectory-level** rewards, making it hard to dissect the effect of good/bad responses inside a conversation.
>
> - In depth, MRs answers **`"How does the current model response impact future interactions?."`** We train the model to produce responses that maximize the final reward given the context.
> In contrast, MTPO leverages preference rewards between two conversations, answering **`"Which conversation should the model prefer?"`** where multiple responses are entangled and hard to dissect their influences to the entire conversation.
>
> - Borrowing terms from causality literature [2], we refer to the first mechanism as **interventional**, and the second as **observational**. The interventional mechanism by MRs offers more fine-grained estimation of the effect of model responses.
>
> Therefore, to your questions, the difference is not simply from the user simulator. In fact, the existing methods like MTPO can have user simulators while still relying on post-hoc comparison between conversations. In terms of data creation, the data we generated (data available in the next answer) comes from turn-by-turn filtering guided by MRs, while MTPO is trained on pairs of good and bad conversations.
>
> We hope the explanations convey the distinctions clearly. We have added them to the related works section to improve the paper's quality, as suggested by the reviewer.
>
> ---
>
> ## **[Claims And Evidence] Request to access the multiturn datasets**
>
> We are happy to provide the datasets!
>
> You can click the anonymous link: https://anonymous.4open.science/r/collab-llm/notebooks/load_conv_data.ipynb, we provide the script that loads the full datasets from `data/`. The notebook displays random samples of the data, and can be updated to show different samples. We hope this helps with getting a clearer picture of our dataset!
>
> ---
>
> ## **[Other Suggestions] "the design must be better conveyed to show that this work goes beyond re-designing the self-training framework"**
>
> Thanks for raising the insight about self-training and our work!
>
> Self-training typically involves generating synthetic data to improve model performance. Under this scope, our work indeed 1) conducts synthetic data generation, and 2) leverages this data to improve the model.
>
> However, there are many possible ways our current design can end up being. In particular, our key intuition in (2) is that we want the model to **be aware of future outcomes and prioritize responses with higher long-term impact**, which constitutes our main novelty. To achieve this, we consider both extrinsic and intrinsic metrics for a more user-centric estimation of long-term effects. The rest of the components in (1) are fundamentally aimed at **computing long-term effects**, with the use of a user simulator for forward sampling being just one, albeit minor, contribution of ours. Moreover, our models are not merely trained on offline synthetic data; they also leverage MRs for online training to adapt model behavior.
>
> ---
>
> ## **Summary**
> We hope our responses address your concerns on clarity and data accessibility.
>
> We acknowledge the previous draft may have been concise in explaining differences with related works. We have revised thoroughly and added a paragraph in Related Work discussing the connection with self-training.
>
> We sincerely appreciate your reconsideration of our work in light of our responses. Thank you for your insights!
>
> ---
>
> ## **Reference**
>
> [1] Shani et al. Multi-turn reinforcement learning from preference human feedback. arXiv:2405.14655.
>
> [2] Pearl et al. Causal Inference in Statistics: A Primer. 2016.

---

### Official Review · Reviewer_Fu3N · 2025-03-20

**Overall Recommendation:** 4

**Summary:**

This paper introduces CollabLLM, a training framework designed to enhance the capability of large language models (LLMs) to collaborate with humans in multi-turn interactions.

The basic idea is to introduce forward-looking behaviors in LLMs to maximize long-term collaborative outcomes. This is achieved through a collaborative simulation module, which samples potential future user interactions to assess the impact of current responses using a new metric called Multiturn-aware Reward (MR). The MR combines both extrinsic factors, such as successful task completion, and intrinsic factors, like interaction efficiency, to comprehensively evaluate response quality. By applying reinforcement learning methods to optimize responses according to MR, CollabLLM improves models' abilities to proactively engage users, leading to superior collaborative task performance.

The experimental results show the fine-tuned model actively anticipates user needs, poses relevant follow-up questions, generates targeted content, and offers insightful recommendations.

**Claims And Evidence:**

Yes, the claims made in the submission are supported by clear and convincing evidence specifically in Section 5.

**Essential References Not Discussed:**

Some related work not included in the paper:
1. Abdulhai, Marwa, et al. "Lmrl gym: Benchmarks for multi-turn reinforcement learning with language models." arXiv preprint arXiv:2311.18232 (2023).
2. Shani, Lior, et al. "Multi-turn reinforcement learning from preference human feedback." arXiv preprint arXiv:2405.14655 (2024).

**Experimental Designs Or Analyses:**

Multiturn-aware Reward (MR) Ablation (Section 5.1, Figure 4):
- Validity: The ablation study clearly compares immediate reward methods (Helpfulness, Extrinsic, Extrinsic + Intrinsic) and multiturn-aware reward variants (with window sizes w=1,2,3). The controlled experimental setup is sound because it isolates the effect of reward design clearly and directly evaluates their relative effectiveness.
- Issues: No major issues. However, the authors mention briefly the computational costs associated with larger window sizes, but explicit details about these computational trade-offs are sparse. Including more detail on computational overhead might enhance the practical interpretability of the findings.

Generalization Tests (Section 5.3, Table 2):
- Validity: The authors clearly test model generalization by evaluating on Abg-CoQA, a dataset distinct from training domains, thereby assessing whether learned collaborative strategies transfer effectively.
- Issues: Generalization tests were limited to a single additional dataset. Including multiple diverse external benchmarks could strengthen claims about generalizability.

**Methods And Evaluation Criteria:**

Yes, the proposed methods and evaluation criteria in this paper make sense for addressing the multiturn collaboration problem. Specifically:

The authors clearly identify the limitations of existing methods (like RLHF), which typically reward single-turn responses and do not effectively address long-term user interactions. The introduction of Multiturn-aware Reward (MR) and a collaborative simulation module effectively addresses these limitations by explicitly modeling forward-looking behaviors, thereby ensuring LLMs actively engage in clarifying user intent, leading to better long-term outcomes.

The use of both extrinsic (task-specific success) and intrinsic (interaction efficiency and interactivity) evaluation metrics provides a comprehensive and meaningful assessment of collaboration quality.

**Other Comments Or Suggestions:**

As suggested in strengths and weaknesses.

**Other Strengths And Weaknesses:**

Strengths:
- The paper introduces CollabLLM, a novel training framework designed to enhance multiturn human-LLM collaboration.
- The development of Multiturn-aware Rewards (MR) represents a significant advancement over traditional single-turn reward mechanisms, addressing the limitations of models like RLHF in long-term interactions.
- The paper is well-structured, with clear explanations of the methodology, experimental setups, and results.

Weaknesses:
- While the paper introduces innovative concepts, it does not discuss certain related works that have explored similar themes, such as multi-turn reinforcement learning benchmarks and proactive clarification in language models. Incorporating these references could provide a more comprehensive context for the contributions.
- The paper could benefit from a more detailed discussion of the computational overhead associated with the proposed methods, particularly regarding the scalability of Multiturn-aware Rewards. This information would be valuable for practitioners considering the implementation of CollabLLM.

**Questions For Authors:**

- How does CollabLLM integrate with existing reinforcement learning frameworks and algorithms? Are there specific modifications or considerations required to implement MR within standard RL pipelines? (Clarifying the integration process would help us understand the feasibility of adopting CollabLLM in other systems. Complex integration requirements could pose barriers to implementation.)

**Relation To Broader Scientific Literature:**

1. Addressing Limitations of Single-turn Reward Methods (e.g., RLHF)
- In prior work, RLHF significantly advanced LLM fine-tuning using single-turn feedback, optimizing immediate next-turn responses. This method is now a standard baseline but is inherently limited for multi-turn interactions because it neglects the cumulative effects of model responses on long-term user goals.
- CollabLLM introduces Multiturn-aware Rewards (MR), explicitly modeling the long-term trajectory of human-model interactions.
- Unlike traditional RLHF, MR leverages forward sampling to anticipate conversational impact, thus directly overcoming RLHF’s known limitations regarding long-term interaction quality.

2. Proactive, Clarification-based Interactions
- Prior work explored using LLMs proactively, especially for clarification questions. However, these methods often rely heavily on predefined interaction patterns or specific domains, limiting adaptability. Prompting-based methods attempt similar proactive strategies but struggle with generalizability across diverse user scenarios.
- CollabLLM generalizes proactive collaboration through reinforcement learning, enabling more versatile interactions that dynamically adapt to user intent across different tasks and scenarios.

**Theoretical Claims:**

The paper does not contain explicit theoretical proofs.

---

> ### Author Rebuttal · Authors · 2025-03-31
>
> Thank you for your constructive and thoughtful suggestions touching on the practicalness of our work! We address your remaining concerns:
>
>
> ---
>
> ## **[Experimental Analyses | Weaknesses] "(What is) the computational costs associated with larger window sizes”**
>
> Thanks for this suggestion! In online training, the computational overhead in forward sampling comes from 1) generation from the policy model and 2) generation from the user simulator.
>
> - For (1), the computational overhead and cost are fairly low since we have integrated vLLM [1] in model inference.
> - For (2), we use gpt-4o-mini as the user simulator where we expect user responses to be concise, i.e., the number of output tokens to be small.
>
> We compute the average statistics over 100 future conversations when $w=1,2,3$ on MediumDocEdit-Chat, the document editing task, which has the maximum computational overhead among the three tasks. Please see the table in https://anonymous.4open.science/r/collab-llm/images/cost.png
>
> We have added this information to `Appendix C.3: Computational Cost During Training`, and hopefully this provides clear details.
>
> ---
>
> ## **[Experimental Designs | Weaknesses] "Including multiple diverse external benchmarks could strengthen claims about generalizability.”**
>
> In addition to Adb-CoQA, the model's generalizability is also validated in our user study, where the deployed model is trained on the MediumDocEdit-Chat task with document type restricted to a collection of medium blogs. While in deployment, the model also interacts with users to write personal statements and conduct creative writing.
>
> ---
>
> ## **[Essential References | Weaknesses] Adding related works**
>
> Thanks for providing more references!
>
> The second paper [3] that the reviewer listed is discussed in `Line 631 (Appendix A)` and `Table 4 (Appendix B)`. A difference between CollabLLM and MTPO [4] is that our Multiturn-aware Rewards provide **accurate and turn-level** signals as the long-term effect of each model response, while MTPO applies **trajectory-level** rewards when training the LLMs, making it hard to dissect the effect of good/bad responses inside a trajectory/conversation.
>
> We further added the benchmark paper [2] to `Line 682 (Related Work)`, which will be included in the main paper in the final version. Here is the modified content:
>
> ```Recent benchmarks~\cite{LMRL,MTEval} evaluate LLMs' performance in multiturn settings, measuring the goal orientation and planning capabilities of models across interactions. Several studies...```
>
> We hope this addresses your concern about the comprehensiveness of our related work.
>
> ---
>
>
> ## **[Questions] How does CollabLLM integrate with RL frameworks and what modifications are needed?**
>
> CollabLLM is plug-and-play with two user-defined modifications:
>
>
> 1) **(Optional) User simulator prompt**. We provide the default prompt (the one we used in the paper), while in some cases, the user characteristics can be known ahead of time in certain tasks. For example, consider an LLM for education where users may be students with basic physics understanding. Brief instructions for role-playing can better approximate real conversations.
>
>
> 2) **Metrics.** The intrinsic rewards (interactivity and efficiency) should be applicable to most applications, while users can define other task-specific metrics, such as accuracy, correctness, or, for example, bargaining advantage in debating or deal-making tasks.
>
>
> We are glad that the reviewer raises this question. Our implementation goal is to make CollabLLM easy to use, able to accommodate user-customized tasks, and efficient with a fast inference infrastructure such as vLLM, so as to be compatible with RL training libraries such as [4].
>
>
> ---
>
> ## **Summary**
>
> We thank you for the extensive review and insights! We hope our answers address your concerns on 1) computational overhead, 2) generalization, and 3) related works. We further provide clarification about how to integrate the existing RL training with CollabLLM.
>
> Finally, we greatly appreciate your support, which strengthens the paper in terms of its practical potential. Thank you again!
>
> ---
>
> ## **Reference**
>
> [1] Kwon et al. Efficient Memory Management for Large Language Model Serving with PagedAttention. SOSP 2023.
>
> [2] Abdulhai, Marwa, et al. Lmrl gym: Benchmarks for multi-turn reinforcement learning with language models. arXiv preprint arXiv:2311.18232 (2023).
>
> [3] Shani, Lior, et al. Multi-turn reinforcement learning from preference human feedback. arXiv preprint arXiv:2405.14655 (2024).
>
> [4] von Werra et al. TRL: Transformer Reinforcement Learning. 2020.

---

### Decision · Program_Chairs · 2025-05-01

**Decision:**

Accept (oral)

**Comment:**

This paper addresses an essential problem: improving the LLMs' multi-turn conversational capability. The key strengths of this paper as identified by the reviewers include a method for generating data with a user simulator and multi-turn reward is important and technically sound.  The results are strong, and the authors compare the proposed method against strong baselines. The paper additionally introduces three public benchmarks for multi-turn conversation research.

The authors did substantial work to address all reviewer concerns.